# FrugalML: How to Use ML Prediction APIs More Accurately and Cheaply

**Lingjiao Chen**[1] , **Matei Zaharia** [1], **James Zou**[1,2]
[1]Department of Computer Sciences,   [2] Department of Biomedical Data Science
Stanford University

## Abstract

Prediction APIs offered for a fee are a fast-growing industry and an important part of machine learning as a service. While many such services are available, the heterogeneity in their price and performance makes it challenging for users to decide which API or combination of APIs to use for their own data and budget. We take a first step towards addressing this challenge by proposing FrugalML, a principled framework that jointly learns the strength and weakness of each API on different data, and performs an efficient optimization to automatically identify the best sequential strategy to adaptively use the available APIs within a budget constraint. Our theoretical analysis shows that natural sparsity in the formulation can be leveraged to make FrugalML efficient. We conduct systematic experiments using ML APIs from Google, Microsoft, Amazon, IBM, Baidu and other providers for tasks including facial emotion recognition, sentiment analysis and speech recognition. Across various tasks, FrugalML can achieve up to 90% cost reduction while matching the accuracy of the best single API, or up to 5% better accuracy while matching the best API's cost.

## 1  Introduction

Machine learning as a service (MLaaS) is a rapidly growing industry. For example, one could use Google prediction API [9] to classify an image for $0.0015 or to classify the sentiment of a text passage for $0.00025. MLaaS services are appealing because using such APIs reduces the need to develop one's own ML models. The MLaaS market size was estimated at $1 billion in 2019, and it is expected to grow to $8.4 billion by 2025 [1].

Third-party ML APIs come with their own challenges, however. A major challenge is that different companies charge quite different amounts for similar tasks. For example, for image classification, Face++ charges $0.0005 per image [6], which is 67% cheaper than Google [9], while Microsoft charges $0.0010 [11]. Moreover, the prediction APIs of different providers perform better or worse on different types of inputs. For example, accuracy disparities in gender classification were observed for different skin colors [23, 37]. As we will show later in the paper, these APIs' performance also varies by class—for example, we found that on the FER+ dataset, the Face++ API had the best accuracy on *surprise* images while the Microsoft API had the best performance on *neutral* images. The more expensive APIs are not uniformly better; and APIs tend to have specific classes of inputs where they perform better than alternatives. This heterogeneity in price and in performance makes it challenging for users to decide which API or combination of APIs to use for their own data and budget.

In this paper, we propose FrugalML, a principled framework to address this challenge. FrugalML jointly learns the strength and weakness of each API on different data, then

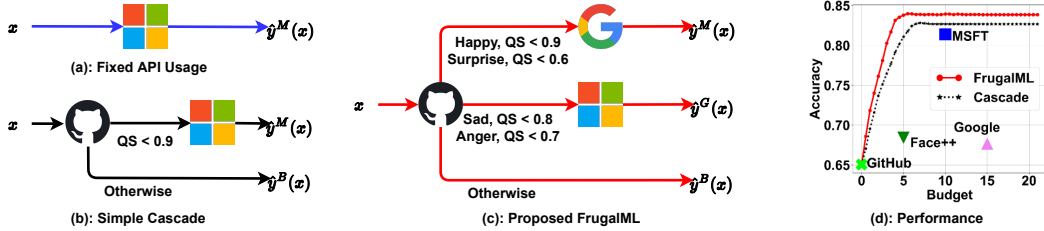

Figure 1: Comparison of different approaches to use ML APIs. Naively calling a fixed API in (a) provides a fixed cost and accuracy. The simple cascade in (b) uses the quality score (QS) from a low-cost open source model to decide whether to call an additional service. Our proposed FrugalML approach, in (c), exploits both the quality score and predicted label to select APIs. Figure (d) shows the benefits of FrugalML on FER+, a facial emotion dataset.

performs an efficient optimization to automatically identify the best adaptive strategy to use all the available APIs given the user's budget constraint. FrugalML leverages the modular nature of APIs by designing adaptive strategies that can call APIs sequentially. For example, we might first send an input to API A. If A returns the label "dog" with high confidence—and we know A tends to be accurate for dogs—then we stop and report "dog". But if A returns "hare" with lower confidence, and we have learned that A is less accurate for "hare," then we might adaptively select a second API B to make additional assessment.

FrugalML optimizes such adaptive strategies to substantially improve prediction performance over simpler approaches such as model cascades with a fixed quality threshold (Figure 1). Through experiments with real commercial ML APIs on diverse tasks, we observe that FrugalML typically reduces costs more than 50% and sometimes up to 90%. Adaptive strategies are challenging to learn and optimize, because the choice of the $2^{nd}$ predictor, if one is chosen, could depend on the prediction and confidence of the first API, and because FrugalML may need to allocate different fractions of its budget to predictions for different classes. We prove that under quite general conditions, there is natural sparsity in this problem that we can leverage to make FrugalML efficient.

**Contributions** To sum up, our contributions are:

1. We formulate and study the problem of learning to optimally use commercial ML APIs given a budget. This is a growing area of importance and is under-explored.

2. We propose FrugalML, a framework that jointly learns the strength and weakness of each API, and performs an optimization to identify the best strategy for using those APIs within a budget constraint. By leveraging natural sparsity in this optimization problem, we design an efficient algorithm to solve it with provable guarantees.

3. We evaluate FrugalML using real-world APIs from diverse providers (e.g., Google, Microsoft, Amazon, and Baidu) for classification tasks including facial emotion recognition, text sentiment analysis, and speech recognition. We find that FrugalML can match the accuracy of the best individual API with up to 90% lower cost, or significantly improve on this accuracy, up to 5%, with the the same cost.

4. We release our code and our dataset[1] of 612,139 samples annotated by commercial APIs as a resource to aid future research in this area.

**Related Work.   MLaaS:** With the growing importance of MLaaS APIs [2, 3, 6, 9, 10, 11], existing research has largely focused on individual API for performance [57], pricing [26], robustness [31], and applications [23, 32, 44]. On the other hand, FrugalML aims at finding strategies to select from or use multiple APIs to reduce costs and increase accuracy.

**Ensemble methods:** A natural approach to exploiting multiple predictors is ensemble methods [25, 29, 45]. While most ensemble methods such as stacking [53], and bagging [22] require predictions from all predictors and thus incur a high cost, mixture of experts

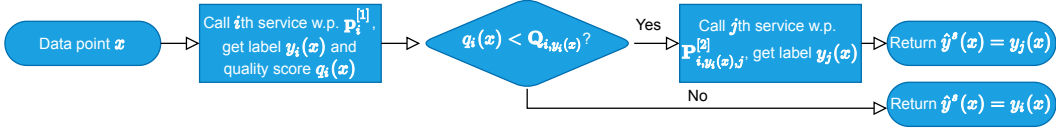

Figure 2: In FrugalML, a base service is first selected and called. If its quality score is smaller than the threshold for its predicted label, FrugalML chooses an add-on service to invoke and returns its prediction. Otherwise, the base service's prediction is returned.

[35, 34, 58] uses gate functions to select one expert (predictor) per data point and is less expensive. Substantial research has focused on developing gate function models, such as SVMs [27, 56], Gaussian Process [28, 55], and neutral networks [47, 46]. However, applying mixture of experts for MLaaS would result in fixed cost and would not allow users to specify a budget as in FrugalML. As we will show later, sometimes FrugalML with a budget constraint can even outperform mixture of experts algorithms while using less budget.

**Model Cascades:** Cascades consisting of a sequence of models are useful to balance the quality and runtime of inference [49, 50, 24, 36, 48, 51, 54, 38]. While model cascades use predicted quality score *alone* to avoid calling computationally expensive models, FrugalML' strategies can utilize both quality score and predicted class to select a downstream expensive add-on service. Designing such strategies requires solving a significantly harder optimization problem, e.g., choosing how to divide the available budget between classes (§3), but also improves performance substantially over using the quality score alone (§4).

## 2 Preliminaries

**Notation.** In our exposition, we denote matrices and vectors in bold, and scalars, sets, and functions in standard script. We let $\mathbf{1}_m$ denote the $m \times 1$ all ones vector, while $\mathbf{1}_{n \times m}$ denotes the all ones $n \times m$ matrix. We define $\mathbf{0}_m, \mathbf{0}_{n \times m}$ analogously. The subscripts are omitted when clear from context. Given a matrix $\mathbf{A} \in \mathbb{R}^{n \times m}$, we let $\mathbf{A}_{i,j}$ denote its entry at location $(i, j)$, $\mathbf{A}_{i,\cdot} \in \mathbb{R}^{1 \times m}$ denote its $i$th row, and $\mathbf{A}_{\cdot,j} \in \mathbb{R}^{n \times 1}$ denote its $j$th column. Let $[n]$ denote $\{1, 2, \cdots, n\}$. Let $\mathbb{1}$ represent the indicator function.

**ML Tasks.** Throughout this paper, we focus on (multiclass) classification tasks, where the goal is to classify a data point $x$ from a distribution $D$ into $L$ label classes. Many real world ML APIs aim at such tasks, including facial emotion recognition, where $x$ is a face image and label classes are emotions (happy, sad, etc), and text sentiment analysis, where $x$ is a text passage and the label classes are attitude sentiment (either positive or negative).

**MLaaS Market.** Consider a MLaaS market consisting of $K$ different ML services which aim at the same classification task. Taken a data point $x$ as input, the $k$th service returns to the user a predicted label $y_k(x) \in [L]$ and its quality score $q_k(x) \in [0, 1]$, where larger score indicates higher confidence of its prediction. This is typical for many popular APIs. There is also a unit cost associated with each service. Let the vector $\mathbf{c} \in \mathbb{R}^K$ denote the unit cost of all services. Then $\mathbf{c}_k = 0.005$ simply means that users need to pay 0.005 every time they call the $k$th service. We use $y(x)$ to denote $x$'s true label, and let $r^k(x) \triangleq \mathbb{1}_{y_k(x) = y(x)}$ be the reward of using the $k$ service on $x$.

## 3 FrugalML: a Frugal Approach to Adaptively Leverage ML Services

In this section, we present FrugalML, a formal framework for API calling strategies to obtain accurate and cheap predictions from a MLaaS market. All proofs are left to the supplemental materials. We generalize the scheme in Figure 1 (c) to $K$ ML services and $L$ label classes. Let a tuple $s \triangleq (\mathbf{p}^{[1]}, \mathbf{Q}, \mathbf{P}^{[2]})$ represent a calling strategy produced by FrugalML. Given an input data $x$, FrugalML first calls a *base service*, denoted by $A_s^{[1]}$, which with probability $\mathbf{p}_i^{[1]}$ is the $i$th service and returns quality score $q_i(x)$ and label $y_i(x)$. Let $D_s$ be the indicator

of whether the quality score is smaller than the threshold value $\mathbf{Q}_{i,y_i(x)}$. If $D_s = 1$, then FrugalML invokes an *add-on service*, denoted by $A_s^{[2]}$, with probability $\mathbf{P}_{i,y_i(x),j}^{[2]}$ being the $j$th service and producing $y_j(x)$ as the predicted label $\hat{y}^s(x)$. Otherwise, FrugalML simply returns label $\hat{y}^s(x) = y_i(x)$ from the base service. This process is summarized in Figure 2. Note that the strategy is adaptive: the choice of the add-on API can depend on the predicted label and quality score of the base model.

The set of possible strategies can be parametrized as $S \triangleq \{(\mathbf{p}^{[1]}, \mathbf{Q}, \mathbf{P}^{[2]}) | \mathbf{p}^{[1]} \succcurlyeq \mathbf{0} \in \mathbb{R}^K, \mathbf{1}^T\mathbf{p}^{[1]} = 1, \mathbf{Q} \in \mathbb{R}^{K \times L}, \mathbf{0} \preccurlyeq \mathbf{Q} \preccurlyeq \mathbf{1}, \mathbf{P}^{[2]} \in \mathbb{R}^{K \times L \times K}, \mathbf{P}^{[2]} \succcurlyeq \mathbf{0}, \mathbf{1}^T\mathbf{P}_{k,\ell,\cdot}^{[2]} = 1\}$. Our goal is to choose the optimal strategy $s^*$ that maximizes the expected accuracy while satisfies the user's budget constraint $b$. This is formally stated as below.

**Definition 1.** *Given a user budget $b$, the optimal FrugalML strategy $s^* = (\mathbf{p}^{[1]*}, \mathbf{Q}^*, \mathbf{P}^{[2]*})$ is*

$$s^* \triangleq \arg\max_{s \in S} \mathbb{E}[r^s(x)] \ s.t. \ \mathbb{E}[\eta^{[s]}(x, \mathbf{c})] \leq b, \tag{3.1}$$

*where $r^s(x) \triangleq \mathbb{1}_{\hat{y}^s(x)=y(x)}$ is the reward and $\eta^{[s]}(x, \mathbf{c})$ the total cost of strategy $s$ on $x$.*

**Remark 1.** *The above definition can be generalized to wider settings. For example, instead of 0-1 loss, the reward can be negative square loss to handle regression tasks. We pick the concrete form for demonstration purposes. The cost of strategy $s$, $\eta^{[s]}(x, \mathbf{c})$, is the sum of all services called on $x$. For example, if service 1 and 2 are called for predicting $x$, then $\eta^{[s]}(x, \mathbf{c})$ becomes $\mathbf{c}_1 + \mathbf{c}_2$.*

Given the above formulation, a natural question is how to solve it efficiently. In the following, We first highlight an interesting property of the optimal strategy, *sparsity*, which inspires the design of the efficient solver, and then present the algorithm for the solver.

## 3.1 Sparsity Structure in the Optimal Strategy

We show that if problem 3.1 is feasible and has unique optimal solution, then we must have $\|\mathbf{p}^{[1]*}\| \leq 2$. In other words, the optimal strategy should only choose the base service from at most two services (instead of $K$) in the MLaaS market. This is formally stated in Lemma 1.

**Lemma 1.** *If problem 3.1 is feasible, then there exists one optimal solution $s^* = (\mathbf{p}^{[1]*}, \mathbf{Q}^*, \mathbf{P}^{[2]*})$ such that $\|\mathbf{p}^{[1]*}\| \leq 2$.*

To see this, let us first expand $\mathbb{E}[r^s(x)]$ and $\mathbb{E}[\eta^s(x)]$ by the law of total expectation.

**Lemma 2.** *The expected accuracy is $\mathbb{E}[r^s(x)] = \sum_{i=1}^K \Pr[A_s^{[1]} = i] \Pr[D_s = 0 | A_s^{[1]} = i] \mathbb{E}[r^i(x) | D_s = 0, A_s^{[1]} = i] + \sum_{i,j=1}^K \Pr[A_s^{[1]} = i] \Pr[D_s = 1 | A_s^{[1]} = i] \Pr[A_s^{[2]} = j | D_s = 1, A_s^{[1]} = i] \mathbb{E}[r^j(x) | D_s = 1, A_s^{[1]} = i])$. The expected cost is $\mathbb{E}[\eta^s(x)] = \sum_{i=1}^K \Pr[A_s^{[1]} = i] \Pr[D_s = 0 | A_s^{[1]} = i] \mathbf{c}_i + \sum_{i,j=1}^K \Pr[A_s^{[1]} = i] \Pr[D_s = 1 | A_s^{[1]} = i] \Pr[A_s^{[2]} = j | D_s = 1, A_s^{[1]} = i] (\mathbf{c}_i + \mathbf{c}_j)$.*

Note that both $\mathbb{E}[r^s(x)]$ and $\mathbb{E}[\eta^s(x)]$ are linear in $\Pr[A_s^{[1]} = i]$, which by definition equals $\mathbf{p}_i^{[1]}$. Thus, fixing $\mathbf{Q}$ and $\mathbf{P}^{[2]}$, problem 3.1 becomes a linear programming in $\mathbf{p}^{[1]}$. Intuitively, the corner points of its feasible region must be 2-sparse, since except $\mathbb{E}[\eta^s(x)] \leq b$ and $\mathbf{1}^T\mathbf{p}^{[1]} \leq 1$, all other constraints ($\mathbf{p}^{[1]} \succcurlyeq \mathbf{0}$) force sparsity. As the optimal solution of a linear programming should be a corner point, $\mathbf{p}^{[2]}$ must also be 2-sparse.

This sparsity structure helps reduce the computational complexity for solving problem 3.1. In fact, the sparsity structure implies problem 3.1 becomes equivalent to a *master problem*

$$\max_{(i_1,i_2,p_1,p_2,b_1,b_2) \in C} p_1 g_{i_1}(b_1/p_1) + p_2 g_{i_2}(b_2/p_2) \ s.t. b_1 + b_2 \leq b \tag{3.2}$$

where $c = \{(i_1, i_2, p_1, p_2, b_1, b_2) | i_1, i_2 \in [K], p_1, p_2 \geq 0, p_1 + p_2 = 1, b_1, b_2 \geq 0\}$, and $g_i(b')$ is the optimal value of the *subproblem*

$$\max_{\mathbf{Q}, \mathbf{P}^{[2]}: s = (\mathbf{e}_i, \mathbf{Q}, \mathbf{P}^{[2]}) \in S} \mathbb{E}[r^s(x)] \ s.t. \ \mathbb{E}[\eta^s(x)] \leq b' \tag{3.3}$$

Here, the master problem decides which two services $(i_1, i_2)$ can be the base service, how often $(p_1, p_2)$ they should be invoked, and how large budgets $(b_1, b_2)$ are assigned, while for a fixed base service $i$ and budget $b'$, the subproblem maximizes the expected reward.

## 3.2 A Practical Algorithm

Now we are ready to give the sparsity-inspired algorithm for generating an approximately optimal strategy $\hat{s}$, summarized in Algorithm 1.

---

**Algorithm 1** FrugalML Strategy Training.

---

**Input** : $K, M, \mathbf{c}, b, \{y(x_i), \{q_k(x_i), y_k(x_i)\}_{k=1}^{K}\}_{i=1}^{N}$

**Output:** FrugalML strategy tuple $\hat{s} = \left( \hat{\mathbf{p}}^{[1]}, \hat{\mathbf{Q}}, \hat{\mathbf{P}}^{[2]} \right)$

1: Estimate $\mathbb{E}[r_i(x)|D_s, A_s^{[1]}]$ from the training data $\{y(x_i), \{q_k(x_i), y_k(x_i)\}_{k=1}^{K}\}_{i=1}^{N}$
2: For $i \in [K]$, $b'_m \in [0, \frac{\|2\mathbf{c}\|_{\infty}}{M}, \cdots, \|2\mathbf{c}\|_{\infty}]$, solve problem 3.3 to find optimal value $g_i(b'_m)$
3: For $i \in [K]$, construct function $g_i(\cdot)$ by linear interpolation on $b'_0, b'_1, \cdots, b'_M$.
4: Solve problem 3.2 to find optimal solution $i_1^*, i_2^*, p_1^*, p_2^*, b_1^*, b_2^*$
5: For $t \in [2]$, let $i = i_t^*, b' = b_t^*/p_t^*$, solve problem 3.3 to find the optimal solution $\mathbf{Q}_{[i_t^*]}, \mathbf{P}_{[i_t^*]}^{[2]}$
6: $\hat{\mathbf{p}}^{[1]} = p_1^* \mathbf{e}_{i_1^*} + p_2^* \mathbf{e}_{i_2^*}, \hat{\mathbf{Q}} = \mathbf{Q}_{[i_1^*]} + \mathbf{Q}_{[i_2^*]}, \hat{\mathbf{P}}^{[2]} = \mathbf{P}_{[i_1^*]}^{[2]} + \mathbf{P}_{[i_2^*]}^{[2]}$
7: Return $\hat{s} = \left( \hat{\mathbf{p}}^{[1]}, \hat{\mathbf{Q}}, \hat{\mathbf{P}}^{[2]} \right)$

---

Algorithm 1 consists of three main steps. First, the conditional accuracy $\mathbb{E}[r_i(x)|D_s, A_s^{[i]}]$ is estimated from the training data (line 1). Next (line 2 to line 4), we find the optimal solution $i_1^*, i_2^*, p_1^*, p_2^*, b_1^*, b_2^*$ to problem 3.2. To do so, we first evaluate $g_i(b')$ for $M+1$ different budget values (line 2), and then construct the functions $g_i(\cdot)$ via linear interpolation (line 3) while enforce $g_i(b') = 0, \forall b' \leq \mathbf{c}_i$. Given (piece-wise linear) $g_i(\cdot)$, problem 3.2 can be solved by enumerating a few linear programming (line 4). Finally, the algorithm seeks to find the optimal solution in the original domain of the strategy, by solving subproblem 3.3 for base service being $i_1^*$ and $i_2^*$ separately (line 5), and then align those solutions appropriately (line 6). We leave the details of solving subproblem 3.3 to the supplement material due to space constraint. Theorem 3 provides the performance analysis of Algorithm 1.

**Theorem 3.** *Suppose $\mathbb{E}[r_i(x)|D_s, A_s^{[1]}]$ is Lipschitz continuous with constant $\gamma$ w.r.t. each element in $\mathbf{Q}$. Given $N$ i.i.d. samples $\{y(x_i), \{(y_k(x_i), q_k(x_i))\}_{k=1}^{K}\}_{i=1}^{N}$, the computational cost of Algorithm 1 is $O\left(NMK^2 + K^3M^3L + M^LK^2\right)$. With probability $1 - \epsilon$, the produced strategy $\hat{s}$ satisfies $\mathbb{E}[r^{\hat{s}}(x)] - \mathbb{E}[r^{s^*}(x)] \geq -O\left(\sqrt{\frac{\log \epsilon + \log M + \log K + \log L}{N}} + \frac{\gamma}{M}\right)$, and $\mathbb{E}[\eta^{[\hat{s}]}(x, \mathbf{c})] \leq b$.*

As Theorem 3 suggests, the parameter $M$ is used to balance between computational cost and accuracy drop of $\hat{s}$. For practical cases where $K$ and $L$ (the number of classes) are around ten and $N$ is more than a few thousands, we have found $M = 10$ is a good value for good accuracy and small computational cost. Note that the coefficient of the $K^L$ terms is small: in experiments, we observe it takes only a few seconds for $L = 31, M = 40$. For datasets with very large number of possible labels, we can always cluster those labels into a few "supclasses", or adopt approximation algorithms to reduce $O(M^L)$ to $O(M^2)$ (see details in the supplemental materials). In addition, slight modification of $\hat{s}$ can satisfy *strict budget constraint*: if budgets allows, use $\hat{s}$ to pick APIs; otherwise, switch to the cheapest API.

## 4 Experiments

We compare the accuracy and incurred costs of FrugalML to that of real world ML services for various tasks. Our goal is four-fold: (i) understanding when and why FrugalML can reduce cost without hurting accuracy, (ii) evaluating the cost savings by FrugalML, (iii) investigating the trade-offs between accuracy and cost achieved by FrugalML, and (iv) measuring the effect of training data size on FrugalML's performance.

**Tasks, ML services, and Datasets.** We focus on three common ML tasks in different application domains: facial emotion recognition (*FER*) in computer vision, sentiment analysis

Table 1: ML services used for each task. Price unit: USD/10,000 queries. A publicly available (and thus free) GitHub model is also used per task: a convolutional neural network (CNN) [13] pretrained on FER2013 [30] for *FER* , a rule based tool (Bixin [4] for Chinese and Vader [16, 33] for English ) for *SA*, and a recurrent neural network (DeepSpeech) [14, 19] pretrained on Librispeech [43] for *STT*.

| Tasks | ML service | Price | ML service | Price | ML service | Price |
|-------|-----------|-------|-----------|-------|-----------|-------|
| *FER* | Google Vision [9] | 15 | MS Face [11] | 10 | Face++ [6] | 5 |
| *SA* | Google NLP [7] | 2.5 | AMZN Comp [2] | 0.75 | Baidu NLP [3] | 3.5 |
| *STT* | Google Speech [8] | 60 | MS Speech [12] | 41 | IBM Speech [10] | 25 |

Table 2: Datasets sample size and number of classes.

| Dataset | Size | # Classes | Dataset | Size | # Classes | Tasks |
|---------|------|-----------|---------|------|-----------|-------|
| FER+ [20] | 6358 | 7 | RAFDB [39] | 15339 | 7 | *FER* |
| EXPW [59] | 31510 | 7 | AFFECTNET [42] | 287401 | 7 | |
| YELP [18] | 20000 | 2 | SHOP [15] | 62774 | 2 | *SA* |
| IMDB [41] | 25000 | 2 | WAIMAI [17] | 11987 | 2 | |
| DIGIT [5] | 2000 | 10 | AUDIOMNIST [21] | 30000 | 10 | *STT* |
| FLUENT [40] | 30043 | 31 | COMMAND [52] | 64727 | 31 | |

(*SA*) in natural langauge processing), and speech to text (*STT*) in speech recognition. The ML services used for each task as well as their prices are summarized in Table 1. For each task we also found a small open source model from GitHub, which is much less expensive to execute per data point than the commercial APIs. Table 2 lists the statistics for all the datasets used for different tasks. More details can be found in the supplemental materials.

**Facial Emotion Recognition: A Case Study.** Let us start with facial emotion recognition on the FER+ dataset. We set budget $b = 5$, the price of FACE++, the cheapest API (except the open source CNN model from GitHub) and obtain a FrugalML strategy by training on half of FER+. Figure 3 demonstrates the learned FrugalML strategy. Interestingly, as shown in Figure 3(b), FrugalML's accuracy is higher than that of the best ML service (Microsoft Face), while its cost is much lower. This is because base service's quality score, utilized by FrugalML, is a better signal than raw image to identify if its prediction is trustworthy. Furthermore, the quality score threshold, produced by FrugalML also depends on label predicted by the base service. This flexibility helps to increase accuracy as well as to reduce costs. For example, using a universal threshold $0.86$ leads to misclassfication on Figure 3(f), while $0.93$ causes unnecessary add-on service call on Figure 3 (c).

The learned FrugalML strategy can be interpreted by the varying API accuracy given labels and quality scores produced by the base service. As shown in Figure 4, the GitHub API can achieve the highest accuracy given that its predicted label is happy or surprise. Thus, when prediction is surprise or happy, the base service is sufficient for most of the images and thus quite some budget can be saved.

For comparison, we also train a mixture of experts strategy with a softmax gating network and the majority voting ensemble method. The learned mixture of experts always uses Microsoft API, leading to the same accuracy (81%) and same cost ($10). The accuracy of majority voting on the test data is slightly better at 82%, but substantially worse than the performance of FrugalML using a small budget of $5. Majority vote, and other standard ensemble methods, needs to collect the prediction of all services, resulting in a cost ($30) which is 6 times the cost of FrugalML. Moreover, both mixture of experts and ensemble method require fixed cost, while FrugalML gives the users flexibility to choose a budget.

**Analysis of Cost Savings.** Next, we evaluate how much cost can be saved by FrugalML to reach the highest accuracy produced by a single API on different tasks, to obtain some qualitative sense of FrugalML. As shown in Table 3, FrugalML can typically save more than

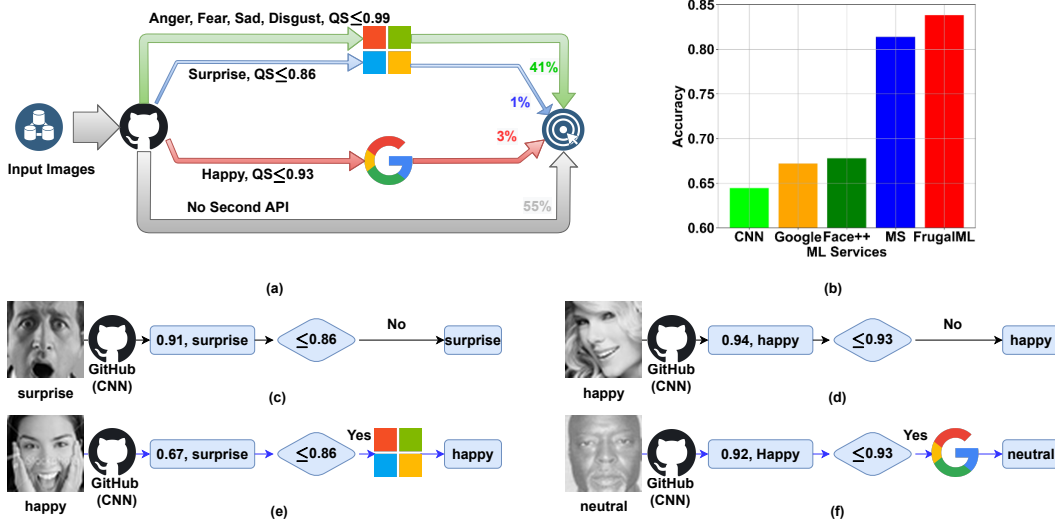

Figure 3: A FrugalML strategy learned on the dataset FER+. (a): data flow. (b): accuracy of all ML services and FrugalML which matches the cost of the cheapest API (FACE++). (c-f): FrugalML prediction process on a few testing data. As shown in (a), on most data (55%), calling the cheap open source CNN from GitHub is sufficient. Thus, FrugalML incurs <50% cost than the most accurate API (Microsoft). Note that unique quality score thresholds for different labels predicted by the base service are learned: e.g., given label, "surprise", 0.86 is used to determine whether (e) or not (c) to call Microsoft, while for label "happy", the learned threshold is 0.93 ((d) and (f)). Such unique thresholds are critical for both accuracy improvement and cost reduction: universally using 0.86 leads to misclassification on (f), while globally adopting 0.93 creates extra cost by called unnecessary add-on service on (c).

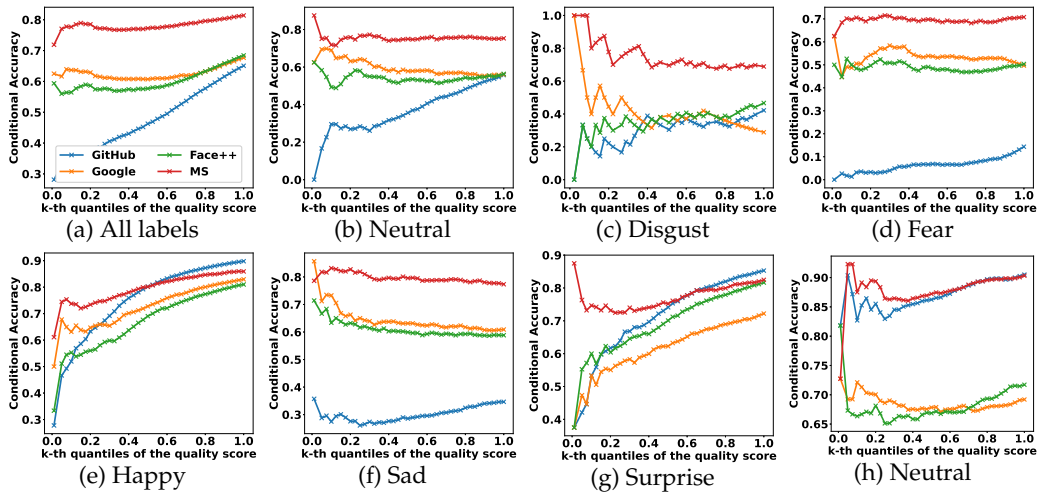

Figure 4: API accuracy on FER+ given labels and quality scores returned by base service.

half of the cost. In fact, the cost savings can be as high as 90% on the AUDIOMNIST dataset. This is likely because the base service's quality score is highly correlated to its prediction accuracy, and thus FrugalML only needs to call expensive services for a few difficult data points. A relatively small saving is reached for *SA* tasks (e.g., on IMDB). This might be that the quality score of the rule based *SA* tool is not highly reliable. Another possible reason is that *SA* task has only two labels (positive and negative), limiting the power of FrugalML.

**Accuracy and Cost Trade-offs.**    Now we dive deeply into the accuracy and cost trade-offs achieved by FrugalML, shown in Figure 5. Here we also compare with two oblations to

Table 3: Cost savings achieved by FrugalML that reaches same accuracy as the best commercial API.

| Dataset | Acc | Price | Cost | Save | Dataset | Acc | Price | Cost | Save |
|---------|-----|-------|------|------|---------|-----|-------|------|------|
| FER+ | 81.4 | 10 | 3.3 | **67%** | RAFDB | 71.7 | 10 | 4.3 | **57%** |
| EXPW | 72.7 | 10 | 5.0 | **50%** | AFFECTNET | 72.2 | 10 | 4.7 | **53%** |
| YELP | 95.7 | 2.5 | 1.9 | **24%** | SHOP | 92.1 | 3.5 | 1.9 | **46%** |
| IMDB | 86.4 | 2.5 | 1.9 | **24%** | WAIMAI | 88.9 | 3.5 | 1.4 | **60%** |
| DIGIT | 82.6 | 41 | 23 | **44%** | COMMAND | 94.6 | 41 | 15 | **63%** |
| FLUENT | 97.5 | 41 | 26 | **37%** | AUDIOMNIST | 98.6 | 41 | 3.9 | **90%** |

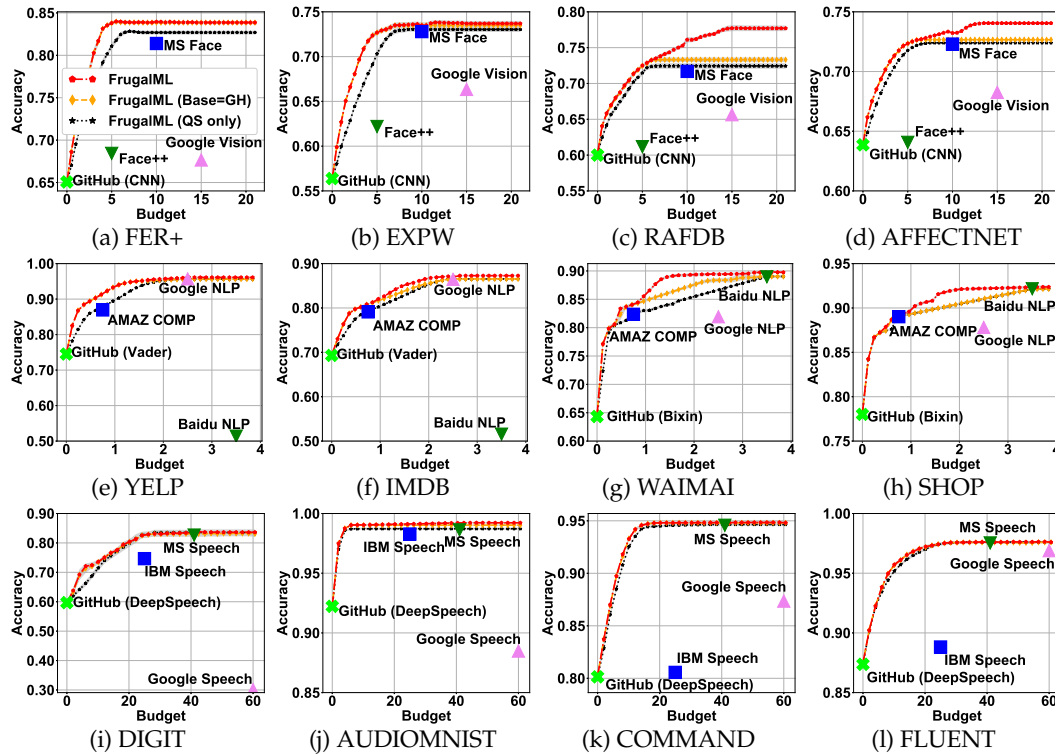

Figure 5: Accuracy cost trade-offs. Base=GH simplifies FrugalML by fixing the free GitHub model as base service , and QS only further uses a universal quality score threshold for all labels. The task of row 1, 2, 3 is *FER*, *SA*, and *STT*, respectively.

FrugalML, "Base=GH", where the base service is forced to be the GitHub model, and "QS only", which further forces a universal quality score threshold across all labels. While using any single ML service incurs a fixed cost, FrugalML allows users to pick any point in its trade-off curve, offering substantial flexibility. In addition to cost saving, FrugalML sometimes can achieve higher accuracy than any ML services it calls. For example, on FER+ and AFFECTNET, more than 2% accuracy improvement can be reached with small cost, and on RAFDB, when a large cost is allowed, more than 5% accuracy improvement is gained. It is also worthy noting that each component in FrugalML helps improve the accuracy. On WAIMAI, for instance, "Base=GH" and "QS only" lead to significant accuracy drops. For speech datasets such as COMMAND, the drop is negligible, as there is no significant accuracy difference between different labels (utterance). Another interesting observation is that there is no universally "best" service for a fixed task. For *SA* task, Baidu NLP achieves the highest accuracy for WAIMAI and SHOP datasets, but Google NLP has best performance on YELP and IMDB. Fortunately, FrugalML adaptively learns the optimal strategy.

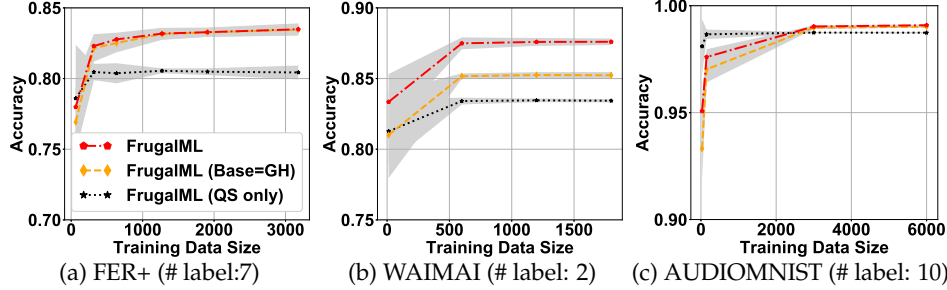

|     |     |     |
| (a) FER+ (# label:7) | (b) WAIMAI (# label: 2) | (c) AUDIOMNIST (# label: 10) |

Figure 6: Testing accuracy v.s.training data size. The fixed budget is 5, 1.2, 20, separately.

**Effects of Training Sample Size** Finally we evaluate how the training sample size affects FrugalML's performance, shown in Figure 6. Note that FrugalML only requires a few thousands training data points for the testing accuracy to converge across all datasets evaluated. This is often more sample-efficient and cost-efficient than training a customized model from scratch. It is also worthy mentioning that larger number of labels usually needs more training samples. For example, 1500 samples might be enough for WAIMAI (#label=2), but 3000 samples are needed for AudioMNIST (#label=10).

## 5 Conclusion and Open Problems

In this work we proposed FrugalML, a formal framework for identifying the best strategy to call ML APIs given a user's budget. Both theoretical analysis and empirical results demonstrate that FrugalML leads to significant cost reduction and accuracy improvement. FrugalML is also efficient to learn: it typically takes a few minutes on a modern machine. Our research characterized the substantial heterogeneity in cost and performance across available ML APIs, which is useful in its own right and also leveraged by FrugalML. Extending FrugalML to produce calling strategies for ML tasks beyond classification (e.g., object detection and language translation) is an interesting future direction. Our discussion with practitioners frequently using ML APIs indicates handling API updates and performance shift is another open problem. As a resource to stimulate further research in MLaaS, we also release a dataset used to develop FrugalML, consisting of 612,139 samples annotated by the APIs, and our code, available at `https://github.com/lchen001/FrugalML`.

## Acknowledgement

This work was supported in part by a Google PhD Fellowship, NSF CCF 1763191, NSF CAREER 1651570 and 1942926, NIH P30AG059307, NIH U01MH098953, grants from the Chan-Zuckerberg Initiative, and affiliate members and other supporters of the Stanford DAWN project—Ant Financial, Facebook, Google, Infosys, NEC, and VMware—as well as Cisco and SAP. We also thank anonymous reviewers for helpful discussion and feedback.

## Potential Broader Impact

ML as a service is a growing industry with substantial economic and societal impact. In this paper, we identify the cost and performance heterogeneity across popular ML APIs, which contributes to the broader understanding of this important but under-explored industry. We proposed a method to automatically reduce user cost while improving accuracy. FrugalML can broadly contribute to the applied ML ecosystem by reducing the expense and complexity of using prediction APIs. This can be a positive impact by increasing accessibility to ML APIs for less well-resourced groups. A potential concern about the ML APIs in general is that they may be trained on biased data and produce biased predictions that could disadvantage certain sub-groups. To tackle this challenge, we are releasing our dataset of over 600k images, text, and utterances that we annotated using commercial APIs. This is a resource for the broad community to use to better understand the biases in existing APIs.

## Footnotes

[1] https://github.com/lchen001/FrugalML

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
