[Supplementary Material]

## A  Extra Notations

Here we introduce a few more notations.

We first let $\cdot, \odot, \otimes$ denote inner, element-wise, and Kronecker product, respectively. Next, Let us introduce a few notations: a matrix $\mathbf{A} \in \mathbb{R}^{K \times L}$, a scalar function $F_{k,\ell}(\cdot) : \mathbb{R} \mapsto \mathbb{R}$ for $k \in [K], \ell \in [L]$, a scalar function $\psi_{k_1,k_2,\ell}(\cdot) : \mathbb{R} \mapsto \mathbb{R}$ for $k_1, k_2 \in [K], \ell \in [L]$, matrix to matrix functions $\mathbf{r}^a(\cdot) : \mathbb{R}^{K \times L} \mapsto \mathbb{R}^{K \times KL}, \mathbf{r}^b(\cdot) : \mathbb{R}^{K \times L} \mapsto \mathbb{R}^{K \times L}$, and $\mathbf{r}^{[-]}(\cdot) : \mathbb{R}^{K \times L} \mapsto \mathbb{R}^{K \times KL}$. $\mathbf{A}$ is given by $\mathbf{A}_{k,\ell} \triangleq \Pr[y_k(x) = \ell]$, which represents the probability of $k$th service producing label $\ell$. The scalar function $F_{k,\ell}(X) \triangleq \Pr[q_k(x) \leq X | y(x) = \ell]$ is the probability of the produced quality score from the $k$th service less than a threshold $X$ conditional on that its predicted label is $\ell$. The scalar function $\psi_{k_1,k_2,\ell}(\cdot)$ is defined as $\psi_{k_1,k_2,\ell}(\alpha) \triangleq \mathbb{E}\left[ r_{k_1}(x) | y_{k_1}(x) = \ell, q_{k_1}(x) \leq F_{k_1,\ell}^{-1}(\alpha) \right]$, i.e., the executed accuracy of the $k_2$ service conditional on that the $k_1$ services produces a label $\ell$ and quality score that is less than $F_{k,\ell}^{-1}(\boldsymbol{\rho}_{k_1,\ell})$. Then those matrix to matrix functions are given by $\mathbf{r}^a_{k_1,K(\ell-1)+k_2}(\boldsymbol{\rho}) \triangleq \psi_{k_1,k_2,\ell}(\boldsymbol{\rho}_{k_1,\ell}), \mathbf{r}^b_{k,\ell}(\boldsymbol{\rho}) \triangleq \psi_{k,k,\ell}(\boldsymbol{\rho}_{k,\ell})$, and $\mathbf{r}^{[-]}(\boldsymbol{\rho}) \triangleq \mathbf{r}^a(\boldsymbol{\rho}) - \mathbf{r}^b(\boldsymbol{\rho}) \otimes \mathbf{1}_K^T$.

## B  Algorithm Subroutines

In this section we provide the details of the subroutines used in the training algorithm for FrugalML. There are in total four components: (i) estimating parameters, (ii) solving subproblem 3.3 to obtain its optimal value and solution, (iii) constructing the function $g_i(\cdot)$, and (iv) solving the master problem 3.2.

**Estimating Parameters.** Instead of directly estimating $\mathbb{E}[r_i(x)|D_s, A_s^{[i]}]$, we estimate $\mathbf{A}, \mathbf{r}^b(\mathbf{1}_{K \times L})$, and $\mathbf{r}^{[-]}(\cdot)$ as defined in Section A, which are sufficient for the subroutines to solve the subproblem 3.3. Let $\hat{\mathbf{A}}, \hat{\mathbf{r}}^b(\mathbf{1}_{K \times L})$, and $\hat{\mathbf{r}}^{[-]}(\cdot)$ be the corresponding estimation from the training datasets. Now we describe how to obtain them from a dataset $\{y(x_i), \{q_k(x_i), y_k(x_i)\}_{k=1}^K\}_{i=1}^N$.

To estimate $\mathbf{A}$, we simply apply the empirical mean estimator and obtain $\hat{\mathbf{A}}_{k,\ell} \triangleq \frac{1}{N} \sum_{i=1}^N \mathbb{1}_{\{y_k(x_i)=\ell\}}$. To estimate $\mathbf{r}^b(\mathbf{1}_{K \times L})$, and $\mathbf{r}^{[-]}(\cdot)$, we first compute $\hat{\psi}_{k_1,k_2,\ell}(\alpha_m) \triangleq \frac{\sum_{i=1}^N \mathbb{1}_{\{y_{k_1}(x_i)=\ell, \hat{q}_{m,k_1,\ell} \geq q_{k_1}(x_i), y_{k_2}(x_i)=y(x_i)\}}}{\sum_{i=1}^N \mathbb{1}_{\{y_{k_1}(x_i)=\ell, \hat{q}_{m,k_1,\ell} \geq q_{k_1}(x_i)\}}}$, for $\alpha_m = \frac{m}{M}, m \in \{0\} \cup [M]$, where $\hat{q}_{m,k,\ell} \triangleq Quantile(\{q_k(x_i)|y_k(x_i) = \ell, i \in [N]\}, \alpha_m)$ is the empirical $\alpha_m$-quantile of the quality score of the $k$th service conditional on its predicted label being $\ell$. Next we estimate $\psi_{k_1,k_2,\ell}(\cdot)$ by linear interpolation, i.e., generating $\hat{\psi}_{k_1,k_2,\ell}(\alpha) \triangleq \frac{\hat{\psi}_{k_1,k_2,\ell}(\alpha_m) - \hat{\psi}_{k_1,k_2,\ell}(\alpha_{m+1})}{\alpha_m - \alpha_{m+1}}(\alpha - \alpha_m) + \hat{\psi}_{k_1,k_2,\ell}(\alpha_m), \alpha \in [\alpha_m, \alpha_{m+1}]$. We can now estimate $\hat{\mathbf{r}}^a_{k_1,K(\ell-1)+k_2}(\boldsymbol{\rho}) \triangleq \psi_{k_1,k_2,\ell}(\boldsymbol{\rho}_{k_1,\ell})$, $\hat{\mathbf{r}}^b_{k,\ell}(\boldsymbol{\rho}) \triangleq \hat{\psi}_{k,k,\ell}(\boldsymbol{\rho}_{k,\ell})$, and finally compute $\hat{\mathbf{r}}^b(\mathbf{1}_{K \times L})_{k,\ell}$ and $\hat{\mathbf{r}}^{[-]}(\boldsymbol{\rho}) \triangleq \hat{\mathbf{r}}^a(\boldsymbol{\rho}) - \hat{\mathbf{r}}^b(\boldsymbol{\rho}) \otimes \mathbf{1}_K^T$.

**Solving subproblem 3.3.** There are 3 steps for solving problem 3.3. First, for $k = i, \ell \in [L]$, invoke Algorithm 2 to compute $\hat{\rho}^{k,\ell}(\beta_m), \hat{\mathbf{\Pi}}^{k,\ell}(\beta_m), \hat{h}_{k,\ell}(\beta_m)$ where $\beta_m = \frac{m}{M}(b' - \mathbf{c}_k), m = 0, 1, \cdots, M$. Next compute $t_1^*, t_2^*, \cdots, t_L^* = \arg\max_{t_1,\cdots,t_L \in [L] \cup \{0\}} \sum_{\ell=1}^L \hat{\mathbf{A}}_{k,\ell} \hat{h}_{k,\ell}(\beta_{t_\ell})$ s.t. $\sum_{\ell=1}^L t_\ell = M$. Finally return $\hat{g}_i(b') \triangleq \sum_{\ell=1}^L \hat{\mathbf{A}}_{k,\ell} \hat{h}_{k,\ell}(\beta_{t_\ell^*})$ as an approximation to the de facto optimal value $g_i(b')$, and the approximately optimal solution $\hat{\mathbf{Q}}_i(b')$ and $\mathbf{P}^{[2]}_i(b')$, where for $\ell \in [L], j \in [K], [\hat{\mathbf{P}}^{[2]}_i(b')]_{i,\ell,j} \triangleq \hat{\mathbf{\Pi}}^{i,\ell}_j(\beta_{t_\ell^*}), [\hat{\mathbf{P}}^{[2]}_i(b')]_{i',\ell,j} \triangleq 0, i' \neq i, [\hat{\mathbf{Q}}_i(b')]_{i,\ell} \triangleq Quantile(\{q_i(x_i)|y_i(x_j) = \ell, j \in [N]\}, \hat{\rho}^{i,\ell}(\beta_{t_\ell^*}))$, and $[\hat{\mathbf{Q}}_i(b')]_{i',\ell} = 0, i' \neq i$.

**Remark 2.** *Algorithm 2 effectively solves the problem*

$$\max_{\rho, \mathbf{\Pi} \in \Omega_2} \hat{\mathbf{r}}^b_{k,\ell}(\mathbf{1}_{K \times L}) + \rho \mathbf{\Pi}^T \cdot \tilde{\mathbf{r}}^{k,\ell}(\rho)$$

$$s.t. \ \rho(\mathbf{\Pi} - \mathbf{\Pi} \odot \mathbf{e}_k)^T \mathbf{c} \leq \beta,$$

$$\text{(B.1)}$$

---

**Algorithm 2** Solver for Problem B.1.

---

**Input** : $\beta, k, \ell, \hat{\mathbf{r}}^b(\mathbf{1}_{K \times L}), \hat{\mathbf{r}}^{[-]}(\cdot)$
**Output:** the optimal solution $\hat{\rho}^{k,\ell}(\beta), \hat{\mathbf{\Pi}}^{k,\ell}(\beta)$, and the optimal value $\hat{h}^{k,\ell}(\beta)$

1: Construct $\tilde{\mathbf{r}}^{kl}(\rho) \triangleq \left[ \hat{\mathbf{r}}^{[-]}_{k,K(\ell-1)+1:K\ell}(\rho \mathbf{1}_{K \times L}) \right]^T$.
2: Construct $\phi_i(\mu) \triangleq \hat{\mathbf{r}}^b_{k,\ell}(\mathbf{1}_{K \times L}) + \min\{\frac{\beta}{\mathbf{c}_i}, \mu\} \tilde{\mathbf{r}}^{k,\ell}_i(\mu)$
3: Construct $\phi_{i,j}(\mu) \triangleq \hat{\mathbf{r}}^b_{k,\ell}(\mathbf{1}_{K \times L}) + \frac{\beta - \mu \mathbf{c}_j}{\mathbf{c}_i - \mathbf{c}_j} \tilde{\mathbf{r}}^{k,\ell}_i(\mu) + \frac{\mu \mathbf{c}_i - \beta}{\mathbf{c}_i - \mathbf{c}_j} \tilde{\mathbf{r}}^{k,\ell}_j(\mu)$
4: Compute $(\mu_1, i_1) = \arg\max_{\mu \in [0,1], i \in [K]} \phi_i(\mu)$
5: Compute $(\mu_2, i_2, j_2) = \arg\max_{\mu \in [\frac{\beta}{\mathbf{c}_i}, \min\{\frac{\beta}{\mathbf{c}_j}, 1\}], i,j \in [K], \mathbf{c}_i > \mathbf{c}_j} \phi_{i,j}(\mu)$.
6: **if** $\phi_{i_1}(\mu_1) \geq \phi_{i_2, j_2}(\mu_2)$ **then**
7:    $\hat{\rho}^{k,\ell}(\beta) = \mu_1, \hat{\mathbf{\Pi}}^{k,\ell}(\beta) = \left[ \mathbb{1}_{\mu_1 < \frac{\beta}{\mathbf{c}_{i_1}}} + \frac{\beta}{\mathbf{c}_i} \mathbb{1}_{\mu_1 \geq \frac{\beta}{\mathbf{c}_{i_1}}} \right] \mathbf{e}_{i_1}, \hat{h}_{k,\ell}(\beta) = \phi_{i_1}(\mu_1)$
8: **else**
9:    $\hat{\rho}^{k,\ell}(\beta) = \mu_2, \hat{\mathbf{\Pi}}^{k,\ell}(\beta) = \frac{\beta/\mu_2 - \mathbf{c}_{j_2}}{\mathbf{c}_{j_2} - \mathbf{c}_{j_2}} \mathbf{e}_{i_2} + \frac{\mathbf{c}_{i_2} - \beta/\mu_2}{\mathbf{c}_{i_2} - \mathbf{c}_{i_2}} \mathbf{e}_{j_2}, \hat{h}_{k,\ell}(\beta) = \phi_{i_1}(\mu_1)$.
   Return $\hat{\rho}^{k,\ell}(\beta), \hat{\mathbf{\Pi}}^{k,\ell}(\beta), \hat{h}_{k,\ell}(\beta)$

---

where $\Omega_2 = \{(\boldsymbol{\rho}, \mathbf{\Pi}) | \boldsymbol{\rho} \in [0,1], \mathbf{\Pi} \in \mathbb{R}^K, \mathbf{0} \preccurlyeq \mathbf{\Pi} \preccurlyeq \mathbf{1}, \mathbf{\Pi}^T \cdot \mathbf{1}_K = 1\}$ and $\tilde{\mathbf{r}}^{kl}(\rho) : \mathbb{R} \mapsto \mathbb{R}^K$ is the transpose of $\hat{\mathbf{r}}^{[-]}_{k,K(\ell-1)+1:K\ell}(\rho \mathbf{1}_{K \times L})$. Observe that the function $\hat{\mathbf{r}}^{[-]}(\cdot)$ by construction is piece wise linear, and thus $\tilde{\mathbf{r}}^{k\ell}(\rho)$ is also piece wise linear. Thus, $\psi_i(\cdot)$ and $\psi_{i,j}(\cdot)$ are piece wise quadratic functions. Thus, the optimization problems in Algorithm 2 (line 4 and line 5) can be efficiently solved, simply by optimizing a quadratic function for each piece.

**Constructing $g_i(\cdot)$.** We construct an approximation to $g_i(\cdot)$, denoted by $\hat{g}^{LI}_i(\cdot)$. The construction is based on linear interpolation using $\hat{g}_i(\theta_m)$ as well as $\hat{g}_i(\mathbf{c}_i)$ which by definition is 0. More precisely, $\hat{g}^{LI}_i(\theta) \triangleq 0, \theta \leq \mathbf{c}_i, \hat{g}^{LI}_i(\theta) \triangleq \frac{\hat{g}_i(\theta_m) - \hat{g}_i(\theta_{m+1})}{\theta_m - \theta_{m+1}} (\theta - \theta_{m+1}) + \hat{g}_i(\theta_{m+1}), \theta_{m+1} \geq \theta \geq \theta_m \geq \mathbf{c}_i$, and $\hat{g}^{LI}_i(\theta) \triangleq \frac{\hat{g}_i(\theta_m)}{\theta_m} \theta, \theta_m \geq \theta \geq \mathbf{c}_i \geq \theta_{m-1}$. Here, $\theta_m \triangleq b'_m = \frac{\|2\mathbf{c}\|_\infty}{M}$.

**Solving Master Problem 3.2.** To solve Problem 3.2, let us first denote $\Omega_3 = \{\mathbf{x} \in \mathbb{R}^4 | x \succcurlyeq 0, \mathbf{x}_1 + \mathbf{x}_2 = 1\}$ and $\Omega_{3,m_1,m_2} \triangleq \{\mathbf{x} \in \Omega_3 | \theta_{m_i-1}\mathbf{x}_i \leq \mathbf{x}_{i+3} \leq \theta_{m_i}\mathbf{x}_{i+3}, i = 1,2\}$, for $m_1 \in [M], m_2 \in [M]$. For each $i_1, i_2, m_1, m_2$, first compute $\hat{g}^\Sigma(i_1, i_2, m_1, m_2) \triangleq \max_{(p_1, p_2, b_1, b_2) \in \Omega_{3,m_1,m_2}} p_1 \hat{g}^{LI}_{i_1}(b_1/p_1) + p_2 \hat{g}^{LI}_{i_2}(b_2/p_2)$ s.t. $b_1 + b_2 = b$, a linear programming by construction. Next compute $i^*_1, i^*_2, m^*_1, m^*_2 \triangleq \arg\max_{i_1, i_2, m_1, m_2} \hat{g}^\Sigma(i_1, i_2, m_1, m_2)$ and $(p^*_1, p^*_2, b^*_1, b^*_2) \triangleq \max_{(p_1, p_2, b_1, b_2) \in \Omega_{3,m^*_1,m^*_2}} p_1 \hat{g}^{LI}_{i^*_1}(b_1/p_1) + p_2 \hat{g}^{LI}_{i^*_2}(b_2/p_2)$ s.t. $b_1 + b_2 = b$. Finally return the corresponding solution $i^*_1, i^*_2, p^*_1, p^*_2, b^*_1, b^*_2$.

## C  Missing Proofs

### C.1  Helpful Lemmas

We first provide some useful lemmas throughout this section.

**Lemma 4.** *Suppose the linear optimization problem*

$$\max_{\mathbf{z} \in \mathbb{R}^K} \mathbf{u}^T \mathbf{z}$$
$$s.t. \mathbf{v}^T \mathbf{z} \leq w, \mathbf{1}^T \mathbf{z} \leq 1, \mathbf{z} \geq 0$$

*is feasible. Then there exists one optimal solution $\mathbf{z}^*$ such that $\|\mathbf{z}^*\|_0 \leq 2$.*

*Proof.* Let $\mathbf{z}^*$ be one solution. If $\|\mathbf{z}^*\|_0 \leq 2$, then the statement holds. Suppose $\|\mathbf{z}^*\|_0 = nnz > 2$ (and thus $K \geq 3$). W.l.o.g., let the first *nnz* elements in $\mathbf{z}^*$ be the nonzero elements. Let

$i_{\min} = \arg\min_{i:i\leq nnz} \mathbf{v}_i$ and $i_{\max} = \arg\max_{i:i\leq nnz} \mathbf{v}_i$. If $\mathbf{v}_{i_{\max}} > \mathbf{v}_{i_{\min}}$, construct $\mathbf{z}'$ by

$$
\mathbf{z}'_i = \begin{cases}
\mathbf{z}^*_i(=0), & \text{if } i > nnz \\
\frac{\mathbf{v}^T_{1:nnz}\mathbf{z}^*_{1:nnz} - \mathbf{v}_{i_{\min}}\mathbf{1}^T\mathbf{z}_{1:nnz}}{\mathbf{v}_{i_{\max}} - \mathbf{v}_{i_{\min}}}, & \text{if } i = i_{\max} \\
\frac{-\mathbf{v}^T_{1:nnz}\mathbf{z}^*_{1:nnz} + \mathbf{v}_{i_{\max}}\mathbf{1}^T\mathbf{z}_{1:nnz}}{\mathbf{v}_{i_{\max}} - \mathbf{v}_{i_{\min}}}, & \text{if } i = i_{\min} \\
0, & \text{otherwise}
\end{cases}
$$

Otherwise, construct $\mathbf{z}'$ by

$$
\mathbf{z}'_i = \begin{cases}
\mathbf{z}^*_i(=0), & \text{if } i > nnz \\
\mathbf{1}^T\mathbf{z}^*_{1:nnz}, & \text{if } i = i_{\max} \\
0, & \text{otherwise}
\end{cases}
$$

Now our goal is to prove that $\mathbf{z}'$ is one optimal solution and $\|\mathbf{z}'\|_0 \leq 2$.

(i) We first show that $\mathbf{z}'$ is a feasible solution.

(1) $\mathbf{v}_{i_{\max}} > \mathbf{v}_{i_{\min}}$: If $i \notin \{i_{\max}, i_{\min}\}$, clearly $\mathbf{z}'_i = 0 \geq 0$. Since $\mathbf{z}^*$ is feasible, $\mathbf{z}^*_{1:nnz} \geq 0$. By definition, $\mathbf{z}'_{i_{\max}} = \frac{\mathbf{v}^T_{1:nnz}\mathbf{z}^*_{1:nnz} - \mathbf{v}_{i_{\min}}\mathbf{1}^T\mathbf{z}_{1:nnz}}{\mathbf{v}_{i_{\max}} - \mathbf{v}_{i_{\min}}} = \frac{1}{\mathbf{v}_{i_{\max}} - \mathbf{v}_{i_{\min}}}\sum_{j=1}^{nnz}(\mathbf{v}_j - \mathbf{v}_{i_{\min}})\mathbf{z}^*_j \geq 0$, and similarly $\mathbf{z}'_{i_{\min}} \geq 0$. Thus, we have $\mathbf{z}' \geq 0$.

In addition,

$$
\mathbf{v}^T\mathbf{z}' = \frac{\mathbf{v}^T_{1:nnz}\mathbf{z}^*_{1:nnz} - \mathbf{v}_{i_{\min}}\mathbf{1}^T\mathbf{z}_{1:nnz}}{\mathbf{v}_{i_{\max}} - \mathbf{v}_{i_{\min}}}\mathbf{v}_{i_{\max}} + \frac{-\mathbf{v}^T_{1:nnz}\mathbf{z}^*_{1:nnz} + \mathbf{v}_{i_{\max}}\mathbf{1}^T\mathbf{z}_{1:nnz}}{\mathbf{v}_{i_{\max}} - \mathbf{v}_{i_{\min}}}\mathbf{v}_{i_{min}}
$$
$$
= \mathbf{v}^T_{1:nnz}\mathbf{z}^*_{1:nnz} = \mathbf{v}^T\mathbf{z}^* \leq w
$$

where the last equality is due to the fact that $\mathbf{z}_i = 0, \forall i > nnz$. Similiarly, we have

$$
\mathbf{1}^T\mathbf{z}' = \frac{\mathbf{v}^T_{1:nnz}\mathbf{z}^*_{1:nnz} - \mathbf{v}_{i_{\min}}\mathbf{1}^T\mathbf{z}_{1:nnz}}{\mathbf{v}_{i_{\max}} - \mathbf{v}_{i_{\min}}} + \frac{-\mathbf{v}^T_{1:nnz}\mathbf{z}^*_{1:nnz} + \mathbf{v}_{i_{\max}}\mathbf{1}^T\mathbf{z}_{1:nnz}}{\mathbf{v}_{i_{\max}} - \mathbf{v}_{i_{\min}}}
$$
$$
= \mathbf{1}^T_{1:nnz}\mathbf{z}^*_{1:nnz} = \mathbf{1}^T\mathbf{z}^* \leq 1.
$$

(2) $\mathbf{v}_{i_{\max}} = \mathbf{v}_{i_{\min}}$: It is clear that $\mathbf{z}' \geq 0$ and $\mathbf{1}^T\mathbf{z} \leq 1$ by definition. Note that by definition $\mathbf{v}^T\mathbf{z}' = \mathbf{v}_{i_{\max}}\mathbf{1}^T\mathbf{z}^*_{1:nnz}$. $\mathbf{v}_{i_{\max}} = \mathbf{v}_{i_{\min}}$ implies that for $i = 1, 2, \cdots, nnz, \mathbf{v}_i = \mathbf{v}_{i_{\max}}$, and thus $\mathbf{v}_{i_{\max}}\mathbf{1}^T\mathbf{z}^*_{1:nnz} = \mathbf{v}^T_{1:nnz}\mathbf{z}^*_{1:nnz}$. Note that only the first $nnz$ elements in $\mathbf{z}^*$ are nonzeros, we have $\mathbf{v}^T_{1:nnz}\mathbf{z}^*_{1:nnz} = \mathbf{v}^T\mathbf{z}^*$. That is to say,

$$
\mathbf{v}^T\mathbf{z}' = \mathbf{v}^T\mathbf{z}^* \leq w
$$

Hence, we have shown that $\mathbf{v}^T\mathbf{z}' \leq w, \mathbf{1}^T\mathbf{z}' \leq 1, \mathbf{z}' \geq 0$ always hold, i.e., $\mathbf{z}'$ is a feasible solution to the linear optimization problem.

(ii) Now we show that $\mathbf{z}'$ is one optimal solution, i.e., $\mathbf{u}^T\mathbf{z}' = \mathbf{u}^T\mathbf{z}^*$.

The Lagrangian function of the linear optimization problem is

$$
\mathcal{L}(\mathbf{z}, \boldsymbol{\mu}) = \mathbf{u}^T\mathbf{z} + \boldsymbol{\mu}_1(\mathbf{v}^T\mathbf{z} - w) + \boldsymbol{\mu}_2(\mathbf{1}^T\mathbf{z}) + \sum_{i=1}^{K}\boldsymbol{\mu}_{i+2}(-\mathbf{z}_i)
$$

Since $\mathbf{z}^*$ is one optimal solution and clearly LCQ (linearity constraint qualification) is satisfied, KKT conditions must hold. That is, there exists $\boldsymbol{\mu}$ such that

$$
\frac{\partial\mathcal{L}(\mathbf{z}^*, \boldsymbol{\mu})}{\partial\mathbf{z}_i} = \mathbf{u}_i + \boldsymbol{\mu}_1\mathbf{v}_i + \boldsymbol{\mu}_2 - \boldsymbol{\mu}_{i+2} = 0, \forall i
$$
$$
\boldsymbol{\mu}_1(\mathbf{v}^T\mathbf{z}^* - w) = 0, \boldsymbol{\mu}_2(\mathbf{1}^T\mathbf{z}^*) = 0, \boldsymbol{\mu}_{i+2}\mathbf{z}^*_i = 0, \forall i
$$
$$
\boldsymbol{\mu} \geq 0
$$
$$
\mathbf{v}^T\mathbf{z}^* \leq w, \mathbf{1}^T\mathbf{z}^* \leq 1, \mathbf{z} \geq 0
$$

For ease of exposition, denote $\hat{\boldsymbol{\mu}} = [\boldsymbol{\mu}_3, \boldsymbol{\mu}_4, \cdots, \boldsymbol{\mu}_{K+2}]^T$. The first condition implies $\mathbf{u}_i = -\boldsymbol{\mu}_1 \mathbf{v}_i - \boldsymbol{\mu}_2 + \boldsymbol{\mu}_{i+2}, \forall i$, which is equivalent to $\mathbf{u} = -\boldsymbol{\mu}_1 \mathbf{v} - \boldsymbol{\mu}_2 \mathbf{1} + \hat{\boldsymbol{\mu}}$. Thus, we have

$$\mathbf{u}^T \mathbf{z}' = -\boldsymbol{\mu}_1 \mathbf{v}^T \mathbf{z}' - \boldsymbol{\mu}_2 \mathbf{1}^T \mathbf{z}' + \hat{\boldsymbol{\mu}}^T \mathbf{z}'$$

$$\mathbf{u}^T \mathbf{z}^* = -\boldsymbol{\mu}_1 \mathbf{v}^T \mathbf{z}^* - \boldsymbol{\mu}_2 \mathbf{1}^T \mathbf{z}^* + \hat{\boldsymbol{\mu}}^T \mathbf{z}^*$$

The condition $\boldsymbol{\mu}_{i+2} \mathbf{z}_i^* = 0$ implies that at least one of the terms must be 0. Since it holds for every $i$, the summation over $i$ is also 0, i.e., $\hat{\boldsymbol{\mu}}^T \mathbf{z}^* = \sum_{i=1}^{K} \boldsymbol{\mu}_{i+2} \mathbf{z}_i^* = 0$. Noting that the first $nnz$ elements in $\mathbf{z}^*$ are nonzeros, we must have $\boldsymbol{\mu}_{i+2} = 0, i \leq nnz$, and in particular, $\boldsymbol{\mu}_{i_{\max}+2} = \boldsymbol{\mu}_{i_{\min}+2} = 0$. Hence, $\hat{\boldsymbol{\mu}}^T \mathbf{z}' = \sum_{i=1}^{K} \boldsymbol{\mu}_{i+2} \mathbf{z}_i' = \boldsymbol{\mu}_{i_{\max}+2} \mathbf{z}_{i_{\max}}' + \boldsymbol{\mu}_{i_{\min}+2} \mathbf{z}_{i_{\min}}' = 0$. Thus, we have

$$\mathbf{u}^T \mathbf{z}' = -\boldsymbol{\mu}_1 \mathbf{v}^T \mathbf{z}' - \boldsymbol{\mu}_2 \mathbf{1}^T \mathbf{z}'$$

$$\mathbf{u}^T \mathbf{z}^* = -\boldsymbol{\mu}_1 \mathbf{v}^T \mathbf{z}^* - \boldsymbol{\mu}_2 \mathbf{1}^T \mathbf{z}^*$$

In part (i), it is shown that $\mathbf{v}^T \mathbf{z}^* = \mathbf{v}^T \mathbf{z}'$ and $\mathbf{1}^T \mathbf{z}^* = \mathbf{1}^T \mathbf{z}'$. Hence, we must have

$$\mathbf{u}^T \mathbf{z}^* = \mathbf{u}^T \mathbf{z}'$$

In other words, $\mathbf{z}'$ has the same objective function value as $\mathbf{z}^*$. Since $\mathbf{z}^*$ is one optimal solution, $\mathbf{z}'$ must also be one optimal solution (since it is also feasible as shown in part (i)). By definition, $\|\mathbf{z}'\|_0 \leq 2$, which finishes the proof. $\qquad \square$

**Lemma 5.** *Let $F(w)$ be the optimal value of the linear optimization problem*

$$\max_{\mathbf{z} \in \mathbb{R}^K} \mathbf{u}^T \mathbf{z}$$

$$s.t.\ \mathbf{v}^T \mathbf{z} \leq w, \mathbf{z} \geq 0, \mathbf{C}\mathbf{z} \leq \mathbf{d}$$

*where $\mathbf{u}, \mathbf{C}, \mathbf{v} \geq \mathbf{0}$, $\mathbf{d} > 0$. Then $F(w)$ is Lipschitz continuous.*

*Proof.* Note that since $\mathbf{d} > \mathbf{0}$, there exists some $w^*$, such that its corresponding optimal $\mathbf{z}^*$ satisfies $\mathbf{C}\mathbf{z}^* < d$. Thus, $\mathbf{z}^*$ must also be the optimal solution to

$$\max_{\mathbf{z} \in \mathbb{R}^K} \mathbf{u}^T \mathbf{z}$$
$$s.t.\ \mathbf{v}^T \mathbf{z} \leq w^*, \mathbf{z} \geq 0 \tag{C.1}$$

If $\mathbf{v}^T \mathbf{z}^* < w^*$, then $\hat{\mathbf{z}} = \frac{w^*}{\mathbf{v}^T \mathbf{z}} \mathbf{z}^*$ is also a feasible solution, but $\mathbf{u}^T \hat{\mathbf{z}} = \frac{w^*}{\mathbf{v}^T \mathbf{z}} \mathbf{u}^T \mathbf{z}^* > \mathbf{u}^T \mathbf{z}^*$, a contradiction. Thus, we must have $\mathbf{v}^T \mathbf{z}^* = w^*$. Now we claim that $\mathbf{z}' = \frac{w'}{w^*} \mathbf{z}^*$ is one optimal solution to

$$\max_{\mathbf{z} \in \mathbb{R}^K} \mathbf{u}^T \mathbf{z}$$
$$s.t.\ \mathbf{v}^T \mathbf{z} \leq w', \mathbf{z} \geq 0 \tag{C.2}$$

Suppose not. Then there exists another optimal solution $\mathbf{z}''$. Since $\mathbf{z}'$ is not optimal, we must have $\mathbf{u}^T \mathbf{z}'' > \mathbf{u}^T \mathbf{z}' = \frac{w'}{w^*} \mathbf{u}^T \mathbf{z}^*$. Now let $\mathbf{z}''' = \frac{w^*}{w'} \mathbf{z}''$. Then by definition, we must have $\mathbf{z}'''$ is a solution to problem C.1, since $\mathbf{z}''' \geq 0$ and $\mathbf{v}^T \mathbf{z}''' = \mathbf{v}^T \frac{w^*}{w'} \mathbf{z}'' \leq \frac{w^*}{w'} \mathbf{v}^T \mathbf{z}'' \leq \frac{w^*}{w'} w' = w^*$. However, $\mathbf{u}^T \mathbf{z}''' = \mathbf{u}^T \frac{w^*}{w'} \mathbf{z}'' = \frac{w^*}{w'} \mathbf{u}^T \mathbf{z}'' > \frac{w^*}{w'} \mathbf{u}^T \mathbf{z}' = \frac{w^*}{w'} \frac{w'}{w^*} \mathbf{u}^T \mathbf{z}^* = \mathbf{u}^T \mathbf{z}^*$. That is to say, $\mathbf{z}'''$ is a feasible solution to problem C.1 but also have a objective value that is strictly higher than that of the optimal solution. A contradiction.

Thus we must have that $\mathbf{z}'$ is one optimal solution to problem C.2.

Now we consider $0 \leq w' \leq w^*$ and $w' \geq w^*$ separately.

case (i): Suppose $0 \leq w' \leq w^*$. Note that $\mathbf{v}^T \mathbf{z}^* \leq w^*$ since $\mathbf{z}^*$ is a feasible solution to problem C.1 and by assumption $\mathbf{C}\mathbf{z}^* < \mathbf{d}$. Hence, we must have $\mathbf{C}\mathbf{z}' = \frac{w'}{w^*} \mathbf{C}\mathbf{z}' < \mathbf{d}$. That is to say,

adding a constraint $\mathbf{Cz} \leq \mathbf{d}$ to problem C.2 does not change the optimal solution. Thus, $\mathbf{z}'$ is also an optimal solution to

$$\max_{\mathbf{z} \in \mathbb{R}^K} \mathbf{u}^T \mathbf{z}$$
$$s.t. \ \mathbf{v}^T \mathbf{z} \leq w^*, \mathbf{z} \geq 0, \mathbf{Cz} \leq \mathbf{d}$$

Hence, we have $F(w') = \mathbf{u}^T \mathbf{z}' = \mathbf{u}^T \frac{w'}{w^*} \mathbf{z}^* = \frac{w'}{w^*} F(w^*)$. That is to say, if $w' \leq w$, then $F(w')$ is a linear function of $w'$ and thus must be Lipschitz continuous.

case (ii): Suppose $w' \geq w^*$. Note that we have just shown that $\mathbf{z}'$ is one optimal solution to problem C.2. Adding a constraint to problem problem C.2 only leads to smaller objective value. That is to say, $\mathbf{u}^T \mathbf{z}' \geq F(w')$, which is the optimal value to

$$\max_{\mathbf{z} \in \mathbb{R}^K} \mathbf{u}^T \mathbf{z}$$
$$s.t. \ \mathbf{v}^T \mathbf{z} \leq w^*, \mathbf{z} \geq 0, \mathbf{Cz} \leq \mathbf{d}$$

On the other hand, by definition, we have $\mathbf{u}^T \mathbf{z}' = \mathbf{u}^T \frac{w'}{w^*} \mathbf{z}^* = \frac{w'}{w^*} F(w^*)$, and thus we have

$$\frac{w'}{w^*} F(w^*) \geq F(w') \tag{C.3}$$

Now let us consider $w^1 \geq w^1 \geq w^*$. Let $\mathbf{z}^1, \mathbf{z}^2$ be their corresponding solutions. Since $w^2 \geq w^1$, we have $F(w^2) \geq F(w^1)$. Let $\mathbf{z}^3 = \frac{w^1}{w^2} \mathbf{z}^2$. Then $\mathbf{v}^T \mathbf{z}^3 = \mathbf{v}^T \frac{w^1}{w^2} \mathbf{z}^2 \leq \frac{w^1}{w^2} w^2 = w^1$ and $\mathbf{Cz}^3 = \mathbf{C} \frac{w^1}{w^2} \mathbf{z}^2 \leq \frac{w^1}{w^2} \mathbf{d} \leq \mathbf{d}$. That is to say, $\mathbf{z}^3$ is also a solution to

$$\max_{\mathbf{z} \in \mathbb{R}^K} \mathbf{u}^T \mathbf{z}$$
$$s.t. \ \mathbf{v}^T \mathbf{z} \leq w^1, \mathbf{z} \geq 0, \mathbf{Cz} \leq \mathbf{d}$$

Thus, the objective value must be smaller than the optimal one, i.e., $\mathbf{u}^T \mathbf{z}^3 \leq F(w^1)$. Noting that $\mathbf{u}^T \mathbf{z}^3 = \mathbf{u}^T \frac{w^1}{w^2} \mathbf{z}^2 = \frac{w^1}{w^2} F(w^2)$, we have $\frac{w^1}{w^2} F(w^2) \leq F(w^1)$. which is $F(w^2) - F(w^1) \leq \frac{w_2 - w_1}{w_1} F(w^1)$. Note that we have proved $\frac{w'}{w^*} F(w^*) \geq F(w')$ in C.3, i.e., $\frac{F(w')}{w'} \leq \frac{F(w^*)}{w^*}$, for any $w' \geq w^*$. Thus, we have $F(w^1)/w^1 \leq \frac{F(w^*)}{w^*}$. That implies $F(w^2) - F(w^1) \leq \frac{w_2 - w_1}{w_1} F(w^1) \leq (w^2 - w^1) \frac{F(w^*)}{w^*}$. We also have $F(w^1) \leq F(w^2)$. That is to say, for any $w^2 \geq w^1 \geq w^*$, we have $-(w^2 - w^1) \frac{F(w^*)}{w^*} \leq 0 \leq F(w^2) - F(w^1) \leq (w^2 - w^1) \frac{F(w^*)}{w^*}$ and thus we have just proved that $f(w')$ is Lipschitz continuous for $w' \geq w^*$.

Now let us consider all $w$. We have shown that $F(w)$ is Lipschitz continuous when $w \leq w^*$ and when $w \geq w^*$. Let $\gamma_1$ and $\gamma_2$ denote the Lipschitz constant for both case. Now we can prove that $F(w)$ is Lipschitz continuous with constant $\gamma_1 + \gamma_2$ for any $w \geq 0$.

Let us consider any two $w_1, w_2$. If they are both smaller than $w^*$ or larger than $w^*$, then clearly we must have $|F(w_1) - F(w_2)| \leq (\gamma_1 + \gamma_2)|w_1 - w_2|$. We only need to consider when $w_1 \leq w^*$ and $w_2 \geq w^*$. As $F(w)$ is Lipschitz continuous on each side, we have

$$
\begin{aligned}
|F(w_1) - F(w_2)| &= |F(w_1) - F(w^*) + F(w^*) - F(w_2)| \\
&\leq |F(w_1) - F(w^*)| + |F(w^*) - F(w_2)| \\
&\leq \gamma_1 |w_1 - w^*| + \gamma_2 |w_2 - w^*| \\
&\leq \gamma_1 |w_1 - w_2| + \gamma |w_2 - w_1| = (\gamma_1 + \gamma_2)|w_1 - w_2|
\end{aligned}
$$

where the first inequality is by triangle inequality, the second ineuqaltiy is by the Lipschitz continuity of $F(w)$ on each side, and the last inequality is due to the assumption that $w_1 \leq w^*$ and $w_2 \geq w^*$. Thus, we can conclude that $F(w)$ must be Lipschitz continuous on for any $w \geq 0$.

$\square$

**Lemma 6.** *Suppose function $f(x)$ is a Lipschitz continuous with constant $\Delta^1$ on the interval $[a,b]$. Let $x_i = \frac{i}{M}(b-a), i = 0, 1, \cdots, M$. Assume for all $i$, $|\hat{f}(x_i) - f(x_i)| \leq \Delta^2$. Let $\hat{f}^{LI}(x)$ be the linear interpolation using $\hat{f}(x_i)$, i.e., $\hat{f}^{LI}(x) \triangleq \frac{\hat{f}(x_i) - \hat{f}(x_{i-1})}{x_i - x_{i-1}}(x - x_i) + \hat{f}(x_i), x_{i-1} \leq x \leq x_i, \forall i \in [M]$. Then we have $|f(x) - \hat{f}^{LI}(x)| \leq 3\Delta^2 + \frac{2\Delta_1(b-a)}{M}$*

*Proof.* For simplicity, let $\mu = \frac{b-a}{M}$. Suppose $x_{i-1} \leq x \leq x_i$. By construction of $\hat{f}^{LI}(x)$, we must have

$$
\begin{aligned}
|\hat{f}^{LI}(x_i) - \hat{f}^{LI}(x)| &\leq |\hat{f}^{LI}(x_i) - \hat{f}^{LI}(x_{i-1})| = |\hat{f}(x_i) - \hat{f}(x_{i-1})| \\
&= |\hat{f}(x_i) - f(x_i) + f(x_i) - f(x_{i-1}) + f(x_{i-1}) - \hat{f}(x_{i-1})| \\
&\leq |\hat{f}(x_i) - f(x_i)| + |f(x_i) - f(x_{i-1})| + |f(x_{i-1}) - \hat{f}(x_{i-1})| \\
&\leq \Delta^2 + \Delta^1 |x_i - x_{i-1}| + \Delta^2 = 2\Delta^2 + \Delta^1 \mu
\end{aligned}
$$

where the last inequality is due to the Lipschitz continuity and assumption $|\hat{f}(x_i) - f(x_i)| \leq \Delta^2$. Since function $f(x)$ is Lipschitz continuous with constant $\Delta^1$, we have

$$
|f(x) - f(x_i)| \leq \Delta^1 |x - x_i| \leq \Delta^1 |x_i - x_{i-1}| = \Delta^1 \mu
$$

By assumption, we have $|\hat{f}(x_i) - f(x_i)| \leq \Delta^2$.

Combining the above, we have

$$
\begin{aligned}
|f(x) - \hat{f}^{LI}(x)| &= |f(x) - f(x_i) + f(x_i) - \hat{f}^{LI}(x_i) + \hat{f}^{LI}(x_i) - \hat{f}^{LI}(x)| \\
&\leq |f(x) - f(x_i)| + |f(x_i) - \hat{f}^{LI}(x_i)| + |\hat{f}^{LI}(x_i) - \hat{f}^{LI}(x)| \\
&= |f(x) - f(x_i)| + |f(x_i) - \hat{f}(x_i)| + |\hat{f}^{LI}(x_i) - \hat{f}^{LI}(x)| \\
&\leq \Delta^1 \mu + \Delta^2 + 2\Delta^2 + \Delta^1 \mu = 3\Delta^2 + 2\Delta^1 \mu = 3\Delta^2 + \frac{2\Delta_1(b-a)}{M}
\end{aligned}
$$

where the first inequality is due to triangle inequality, and the second inequality is simply applying what we have just shown. Note that this holds regardless of the value of $i$. Thus, this holds for any $x$, which completes the proof. $\qquad\square$

**Lemma 7.** *Let $f, f', g, g'$ be functions defined on $\Omega_{\mathbf{z}}$, such that $\max_{\mathbf{z} \in \Omega_{\mathbf{z}}} |(f\mathbf{z}) - f'(\mathbf{z})| \leq \Delta_1$ and $\max_{\mathbf{z} \in \Omega_{\mathbf{z}}} |g(\mathbf{z}) - g'(\mathbf{z})| \leq \Delta_2$. If*

$$
\mathbf{z}^* = \arg\max_{\mathbf{z} \in \Omega_{\mathbf{z}}} f(\mathbf{z})
$$
$$
s.t. g(\mathbf{z}) \leq 0
$$

*and*

$$
\mathbf{z}' = \arg\max_{\mathbf{z} \in \Omega_{\mathbf{z}}} f'(\mathbf{z})
$$
$$
s.t. g'(\mathbf{z}) \leq \Delta_2,
$$

*then we must have*

$$
f(\mathbf{z}') \geq f(\mathbf{z}^*) - 2\Delta_1
$$
$$
g(\mathbf{z}') \leq 2\Delta_2.
$$

*Proof.* Note that $\max_{\mathbf{z} \in \Omega_{\mathbf{z}}} |(f\mathbf{z}) - f'(\mathbf{z})| \leq \Delta_1$ implies $f(\mathbf{z}) \geq f'(\mathbf{z}) - \Delta_1$ for any $\mathbf{z} \in \Omega_{\mathbf{z}}$. Specifically,

$$
f(\mathbf{z}') \geq f'(\mathbf{z}') - \Delta_1
$$

Noting $\max_{\mathbf{z} \in \Omega_{\mathbf{z}}} |g(\mathbf{z}) - g'(\mathbf{z})| \leq \Delta_2$, we have $g'(\mathbf{z}^*) \leq g(\mathbf{z}^*) + \Delta_2 \leq \Delta_2$, where the last inequality is due to $g(\mathbf{z}^*) \leq 0$ by definition. Since, $\mathbf{z}^*$ is a feasible solution to the second optimization problem, and the optimal value must be no smaller than the value at $\mathbf{z}^*$. That is to say,

$$
f'(\mathbf{z}') \geq f'(\mathbf{z}^*)
$$

Hence we have
$$f(\mathbf{z}') \geq f'(\mathbf{z}') - \Delta_1 \geq f'(\mathbf{z}^*) - \Delta_1$$
In addition, $\max_{\mathbf{z} \in \Omega_{\mathbf{z}}} |(f\mathbf{z}) - f'(\mathbf{z})| \leq \Delta_1$ implies $f'(\mathbf{z}) \geq f(\mathbf{z}) - \Delta_1$ for any $\mathbf{z} \in \Omega_{\mathbf{z}}$. Thus, we have $f'(\mathbf{z}^*) \geq f(\mathbf{z})^* - \Delta_1$ and thus
$$f(\mathbf{z}') \geq f'(\mathbf{z}^*) - \Delta_1 \geq f(\mathbf{z}^*) - 2\Delta_1$$

By $\max_{\mathbf{z} \in \Omega_{\mathbf{z}}} |g(\mathbf{z}) - g'(\mathbf{z})| \leq \Delta_2$, we have $g(\mathbf{z}') \leq g'(\mathbf{z}') + \Delta_2 \leq 2\Delta_2$, where the last inequality is by definition of $\mathbf{z}'$. $\qquad\square$

## C.2 Proof of Lemma 1

*Proof.* Given the expected accuracy and cost provided by Lemma 2, the problem 3.1 becomes a linear programming over $\Pr[A_s^{[1]} = i] = \mathbf{p}_i^{[1]}$, where the constraints are $\mathbf{p}^{[1]} \geq 0, \mathbf{1}^T \mathbf{p}^{[1]} = 1$ and another linear constraint on $\mathbf{p}^{[1]}$. Note that all items in the optimization are positive, and thus changing the constraint to $\mathbf{p}^{[1]} = 1$ to $\mathbf{p}^{[1]} \leq 1$ does not change the optimal solution. Now, given the constraint $\mathbf{p}^{[1]} \geq 0, \mathbf{1}^T \mathbf{p}^{[1]} \leq 1$ and one more constraint on $\mathbf{p}^{[2]}$ for the linear programing problem over $\mathbf{p}^{[1]}$, we can apply Lemma 4, and conclude that there exists an optimal solution where $\|\mathbf{p}^{[1]*}\| \leq 2$. $\qquad\square$

## C.3 Proof of Lemma 2

*Proof.* Let us first consider the expected accuracy. By law of total expectation, we have
$$\mathbb{E}[r^s(x)] = \sum_{i=1}^{K} \Pr[A_s^{[1]} = i]\mathbb{E}[r^s(x)|A_s^{[1]} = i]$$
And we can further expand the conditional expectation by

$\mathbb{E}[r^s(x)|A_s^{[1]} = i]$
$= \Pr[D_s = 0|A_s^{[1]} = i,]\mathbb{E}[r^s(x)|A_s^{[1]} = i, D_s = 0] + \Pr[D_s = 1|A_s^{[1]} = i,]\mathbb{E}[r^s(x)|A_s^{[1]} = i, D_s = 1]$
$= \Pr[D_s = 0|A_s^{[1]} = i,]\mathbb{E}[r^i(x)|A_s^{[1]} = i, D_s = 0] + \Pr[D_s = 1|A_s^{[1]} = i,]\mathbb{E}[r^s(x)|A_s^{[1]} = i, D_s = 1]$

where the last equality is because when $D_s = 0$, i.e., no add-on service is called, the strategy always uses the base service's prediction and thus $r^s(x) = r^i(x)$. For the second term, we can bring in $A_s^{[2]}$ and apply law of total expectation, to obtain

$$\mathbb{E}[r^s(x)|A_s^{[1]} = i, D_s = 1] = \sum_{j=1}^{K} \Pr[A_s^{[2]} = j|A_s^{[1]} = i, D_s = 1]\mathbb{E}[r^s(x)|A_s^{[1]} = i, D_s = 1, A_s^{[2]} = j]$$

$$= \sum_{j=1}^{K} \Pr[A_s^{[2]} = j|A_s^{[1]} = i, D_s = 1]\mathbb{E}[r^j(x)|A_s^{[1]} = i, D_s = 1, A_s^{[2]} = j]$$

where the last equality is by observing that given the second add-on service is $j$, the reward simply becomes $r^j(x)$. Combining the above, we have $\mathbb{E}[r^s(x)] = \sum_{i=1}^{K} \Pr[A_s^{[1]} = i] \Pr[D_s = 0|A_s^{[1]} = i]\mathbb{E}[r^i(x)|D_s = 0, A_s^{[1]} = i] + \sum_{i,j=1}^{K} \Pr[A_s^{[1]} = i] \Pr[D_s = 1|A_s^{[1]} = i] \Pr[A_s^{[2]} = j|D_s = 1, A_s^{[1]} = i]\mathbb{E}[r^j(x)|D_s = 1, A_s^{[1]} = i)]$, which is the desired property.

Similarly, we can expand the expected cost by law of total expectation
$$\mathbb{E}[\eta^s(x)] = \sum_{i=1}^{K} \Pr[A_s^{[1]} = i]\mathbb{E}[\eta^s(x)|A_s^{[1]} = i]$$
And we can further expand the conditional expectation by

$\mathbb{E}[\eta^s(x)|A_s^{[1]} = i]$
$= \Pr[D_s = 0|A_s^{[1]} = i,]\mathbb{E}[\eta^i(x)|A_s^{[1]} = i, D_s = 0] + \Pr[D_s = 1|A_s^{[1]} = i,]\mathbb{E}[\eta^s(x)|A_s^{[1]} = i, D_s = 1]$
$= \Pr[D_s = 0|A_s^{[1]} = i,]\mathbf{c}_i + \Pr[D_s = 1|A_s^{[1]} = i,]\mathbb{E}[\eta^s(x)|A_s^{[1]} = i, D_s = 1]$

where the last equality is because when $D_s = 0$, i.e., no add-on service is called, the strategy always uses the base service's prediction and incurs the base service's cost $\eta^s(x) = \mathbf{c}_i$. For the second term, we can bring in $A_s^{[2]}$ and apply law of total expectation, to obtain

$$\mathbb{E}[\eta^s(x)|A_s^{[1]} = i, D_s = 1] = \sum_{j=1}^{K} \Pr[A_s^{[2]} = j|A_s^{[1]} = i, D_s = 1]\mathbb{E}[\eta^s(x)|A_s^{[1]} = i, D_s = 1, A_s^{[2]} = j]$$

$$= \sum_{j=1}^{K} \Pr[A_s^{[2]} = j|A_s^{[1]} = i, D_s = 1](\mathbf{c}_i + \mathbf{c}_j)$$

where the last equality is because given the base service is $i$ and add-on service is $j$, the cost is simply the sum of their cost $\mathbf{c}_i + \mathbf{c}_j$. Combining all the above equations, we have the expected cost $\mathbb{E}[\eta^s(x)] = \sum_{i=1}^{K} \Pr[A_s^{[1]} = i]\Pr[D_s = 0|A_s^{[1]} = i]\mathbf{c}_i + \sum_{i,j=1}^{K} \Pr[A_s^{[1]} = i]\Pr[D_s = 1|A_s^{[1]} = i]\Pr[A_s^{[2]} = j|D_s = 1, A_s^{[1]} = i](\mathbf{c}_i + \mathbf{c}_j)$, which is the desired term. Thus, we have shown a form of expected accuracy and cost which is exactly the same as in Lemma 2, which completes the proof. $\qquad\square$

### C.4 Proof of Theorem 3

*Proof.* To prove Theorem 3, we need a few new definitions and lemmas, which are stated below.

**Definition 2.** *Let $\hat{\mathbb{E}}[r^s(x)]$ and $\hat{\mathbb{E}}[\eta^{[s]}(x, \mathbf{c})]$ denote the empirically estimated accuracy and cost for the strategy $s$. More precisely, let the empirically estimated accuracy be $\hat{\mathbb{E}}[r^s(x)] \triangleq \sum_{i=1}^{K} \Pr[A_s^{[1]} = i]\hat{\Pr}[D_s = 0|A_s^{[1]} = i]\hat{\mathbb{E}}[r^i(x)|D_s = 0, A_s^{[1]} = i] + \sum_{i,j=1}^{K} \Pr[A_s^{[1]} = i]\hat{\Pr}[D_s = 0|A_s^{[1]} = i]\Pr[A_s^{[2]} = j|D_s = 1, A_s^{[1]} = i]\hat{\mathbb{E}}[r^j(x)|D_s = 1, A_s^{[1]} = i)]$, and the empirically estimated cost be $\hat{\mathbb{E}}[\eta^s(x)] = \sum_{i=1}^{K} \Pr[A_s^{[1]} = i]\hat{\Pr}[D_s = 0|A_s^{[1]} = i]\mathbf{c}_i + \sum_{i,j=1}^{K} \Pr[A_s^{[1]} = i]\hat{\Pr}[D_s = 0|A_s^{[1]} = i]\Pr[A_s^{[2]} = j|D_s = 1, A_s^{[1]} = i](\mathbf{c}_i + \mathbf{c}_j)$.*

**Definition 3.** *Let $s' = (\mathbf{p}^{[1]'}, \mathbf{Q}', \mathbf{P}^{[2]'})$ be the optimal solution to*

$$\max_{s \in S} \hat{\mathbb{E}}[r^s(x)] \, s.t. \, \hat{\mathbb{E}}[\eta^{[s]}(x, \mathbf{c})] \leq b$$

Note that $s^*$ is the optimal strategy, and $s'$ is the optimal strategy when the data distribution is unknown and estimated from $N$ samples, and $\hat{s}$ is the strategy we actually generate with finite computational complexity by Algorithm 1.

The following lemma shows the computational complexity of Algorithm 1.

**Lemma 8.** *The complexity of Algorithm 1 is $O\left(NMK^2 + K^3M^3L + M^LK^2\right)$.*

*Proof.* Estimating **A** requires a pass of all the training data, which gives a $O(NK)$ cost. For each $k_1, k_2, \alpha_m$, we need a pass over training data for the $k_1$th and $k_2$th services to estimate $\psi_{k_1,k_2,\ell}(\alpha_M)$. There are in total $K$ services, and thus this takes $O(NMK^2)$ computational cost.

Algorithm 2 has a complexity of $O(K^2)$. Solving Problem 3.3 invokes Algorithm 2 for each $\ell \in L$ and $m \in [M]$, and thus takes $O(K^2ML)$. Computing $t_i^*$ takes $\binom{M}{L}$, which is $O(M^{L-1})$. That is to say, solving the subproblem 3.3 once requires $O(K^2ML + M^{L-1})$ computational cost. Solving the master problem 3.2 requires invoking the subproblem $MK$ times, where $K$ times stands for each service, and $M$ stands for the linear interpolation. Thus, the total computational cost for optimization process takes $O(K^3M^3LM^LK^2)$. Combining this with the estimation cost $O(NMK^2)$ completes the proof. $\qquad\square$

Next we evaluate how far the estimated accuracy and cost can be from the true expected accuracy and cost for each strategy, which is stated in Lemma 9.

**Lemma 9.** *With probability $1 - \epsilon$, we have for all $s \in S$,*

$$\left| \hat{\mathbb{E}}[r^s(x)] - \mathbb{E}[r^s(x)] \right| \leq O\left( \sqrt{\frac{\log \epsilon + \log M + \log K + \log L}{N}} + \frac{\gamma}{M} \right)$$

$$\left| \hat{\mathbb{E}}[\eta^{[s]}(x, \mathbf{c})] - \mathbb{E}[\eta^{[s]}(x, \mathbf{c})] \right| \leq O\left( \sqrt{\frac{\log \epsilon + \log K + \log L}{N}} \right)$$

(C.4)

*and also*

$$\left| \mathbb{E}[r^{s'}(x)] - \mathbb{E}[r^{s^*}(x)] \right| \leq O\left( \sqrt{\frac{\log \epsilon + \log M + \log K + \log L}{N}} + \frac{\gamma}{M} \right)$$

$$\left| \mathbb{E}[\eta^{[s']}(x, \mathbf{c})] - \mathbb{E}[\eta^{[s^*]}(x, \mathbf{c})] \right| \leq O\left( \sqrt{\frac{\log \epsilon + \log K + \log L}{N}} \right)$$

(C.5)

*Proof.* For each element in $\mathbf{A}$, we simply use a sample mean estimator. Thus, by Chernoff bound, we have $|\hat{\mathbf{A}}_{i,\ell} - \mathbf{A}_{i,\ell}| \geq O(\sqrt{\frac{\log \epsilon}{N}})$ w.p. at most $\epsilon$. For each $\psi_{k_1,k_2,\ell}(\alpha_m)$, we again use a sample mean estimator for the true conditional expected accuracy. We again apply the Chernoff bound, and obtain that for each of $k_1, k_2, \ell, \alpha_m$, $|\psi_{k_1,k_2,\ell}(\alpha_m) - \hat{\psi}_{k_1,k_2,\ell}(\alpha_m)| \geq O(\sqrt{\frac{\log \epsilon}{N}})$ w.p. at most $\epsilon$. Now applying the union bound, we have w.p. $1 - \epsilon$, $|\hat{\mathbf{A}}_{i,\ell} - \mathbf{A}_{i,\ell}| \leq O(\sqrt{\frac{\log \epsilon + \log K + \log L}{N}})$ and $|\psi_{k_1,k_2,\ell}(\alpha_m) - \hat{\psi}_{k_1,k_2,\ell}(\alpha_m)| \leq O(\sqrt{\frac{\log \epsilon + \log M + \log K + \log L}{N}})$, for all $\ell, i, k_1, k_2, m$.

Recall that the function $\hat{\phi}_{k_1,k_2,\ell}(\cdot)$ is estimated by linear interpolation over $\alpha_1, \alpha_2, \cdots, \alpha_M$. By assumption, $\phi_{k_1,k_2,\ell}(\cdot)$ is Lipschitz continuous, and $\alpha \in [0, 1]$. Now applying Lemma 6, we have that the estimated function $\hat{\phi}_{k_1,k_2,\ell}(\cdot)$ cannot be too far away from its true value, i.e.,

$$|\hat{\phi}_{k_1,k_2,\ell}(\alpha) - \phi_{k_1,k_2,\ell}(\alpha)| \leq O(\sqrt{\frac{\log \epsilon + \log M + \log K + \log L}{N}} + \frac{\gamma}{M})$$

Recall the definition $\hat{\mathbf{r}}^a_{k_1,K(\ell-1)+k_2}(\boldsymbol{\rho}) \triangleq \psi_{k_1,k_2,\ell}(\boldsymbol{\rho}_{k_1,\ell})$, $\hat{\mathbf{r}}^b_{k,\ell}(\boldsymbol{\rho}) \triangleq \hat{\psi}_{k,k,\ell}(\boldsymbol{\rho}_{k,\ell})$, and $\hat{\mathbf{r}}^{[-]}(\boldsymbol{\rho}) \triangleq \hat{\mathbf{r}}^a(\boldsymbol{\rho}) - \hat{\mathbf{r}}^b(\boldsymbol{\rho}) \otimes \mathbf{1}_K^T$. Then we know that for each element in those matrix function, its estimated value can be at most $O(\sqrt{\frac{\log \epsilon + \log M + \log K + \log L}{N}} + \frac{\gamma}{M})$ away from its true value. Since the true accuracy is the (weighted) average over those functions, its estimated difference is also $O(\sqrt{\frac{\log \epsilon + \log M + \log K + \log L}{N}} + \frac{\gamma}{M})$. The expected cost can be viewed as a (weighted) average over elements in the matrix $\mathbf{A}_{i,\ell}$, and thus the estimation difference is at most $O(\sqrt{\frac{\log \epsilon + \log K + \log L}{N}})$, which completes the proof.

$\square$

Then we need to bound how much error is incurred due to our computational approximation. In other words, the difference between $s'$ and $\hat{s}$, which is given in Lemma 10.

**Lemma 10.**

$$\left| \hat{\mathbb{E}}[r^{\hat{s}}(x)] - \hat{\mathbb{E}}[r^{s'}(x)] \right| \leq O\left( \frac{\gamma}{M} \right)$$

$$\hat{\mathbb{E}}[\eta^{[\hat{s}]}(x, \mathbf{c})] - \hat{\mathbb{E}}[\eta^{[s']}(x, \mathbf{c})] \leq 0$$

(C.6)

*Proof.* This lemma requires a few steps. The first step is to show that the subroutine to solve subproblem gives a good approximation. Then we can show that subroutine for solving the master problem gives a good approximation. Finally combing those two, we can prove this lemma.

Let us start by showing that the subroutine to solve subproblem gives a good approximation.

**Lemma 11.** *For any $b'$, The subroutine for solving problem 3.3 produces a strategy $s(i, b') \triangleq (\mathbf{e}_i, \hat{\mathbf{Q}}_i(b'), \hat{\mathbf{P}}^{[2]}{}_i(b'))$ with the empirical accuracy $\hat{g}_i(b')$ s.t. the empirical accuracy is within $O(\frac{\gamma L}{M})$ from the optimal, i.e., $|\hat{g}_i(b') - g'_i(b')| \leq O(\frac{\gamma}{M})$, and the cost constraint is satisfied, i.e., $\hat{\mathbb{E}}[\gamma^{s(i,b')(x)}] \leq b'$ .*

*Proof.* This requires two lemmas.

**Lemma 12.** *For any input, Algorithm 2 gives the exact optimal solution and optimal value to problem B.1.*

*Proof.* To prove this lemma, we simply note that the problem B.1 also has a sparse structure, which is stated below.

**Lemma 13.** *For any constant $\eta$, function $\phi(\cdot): \mathbb{R} \mapsto \mathbb{R}^K$, and $\Omega_2 = \{\rho, \mathbf{\Pi} | 0 \leq \rho \leq 1, \mathbf{\Pi}^T \mathbf{1} = 1, \mathbf{\Pi} \succcurlyeq 0\}$. Suppose the following optimization problem*

$$\max_{\rho, \mathbf{\Pi} \in \Omega_2} \eta + \rho \mathbf{\Pi}^T \cdot \phi(\rho)$$

$$\text{s.t. } \rho(\mathbf{\Pi} - \mathbf{\Pi} \odot \mathbf{e}_k)^T \mathbf{c} \leq \beta,$$

*is feasible. Then there exists one optimal solution $(\rho^*, \mathbf{\Pi}^*)$, such that $\mathbf{\Pi}^*$ is sparse and $\|\mathbf{\Pi}^*\|_0 \leq 2$. More specifically, one of the following must hold:*

- $\mathbf{\Pi}_i^* = 1$ *for some $i$, and $\mathbf{\Pi}_{k'}^* = 0$, for all $k' \neq i$*

- $\mathbf{\Pi}_i^* = \frac{\beta}{\rho \mathbf{c}_i}$ *for some $i$, $\mathbf{\Pi}_k^* = 1 - \frac{\beta}{\rho \mathbf{c}_i}$ and $\mathbf{\Pi}_{k'}^* = 0$, for all $k' \notin \{i, k\}$*

- $\mathbf{\Pi}_i^* = \frac{\beta/\rho - \mathbf{c}_j}{\mathbf{c}_i - \mathbf{c}_j}, \mathbf{\Pi}_j^* = \frac{\mathbf{c}_i - \beta/\rho}{\mathbf{c}_i - \mathbf{c}_j}$, *for some distinct $i, j$, and $\mathbf{\Pi}_{k'}^* = 0$, for all $k' \neq i, j$*

*Proof.* Let $(p^*, \mathbf{\Pi}')$ be one solution. Our goal is to show that there exists a solution $(p^*, \mathbf{\Pi}^*)$ which satistfies the above conditions.

(i) $p^* = 0$: the optimal value does not depend on $\mathbf{\Pi}'$, and thus any $(p^*, \mathbf{\Pi})$ is a solution. In particular, $(p^*, \mathbf{\Pi}^*)$ is a solution where $\mathbf{\Pi}^*$ satisfies the first condition in the statement.

(ii) $p^* \neq 0$: According to Lemma 4, the following linear optimization problem

$$\max_{\mathbf{\Pi} \in \mathbb{R}^K} \sum_{i=1}^K \mathbf{\Pi}_i \bar{r}_{i, p^*}$$

$$\text{s.t. } \sum_{i=1}^K \mathbf{c}_i \mathbf{\Pi}_i \leq \frac{B}{p^*}, \sum_{i=1}^K \mathbf{\Pi}_i \leq 1, \mathbf{\Pi}_i \geq 0 \tag{C.7}$$

has a solution $\mathbf{\Pi}^*$ such that $\|\mathbf{\Pi}^*\|_0 \leq 2$.

We first show that $(p^*, \mathbf{\Pi}^*)$ is one optimal solution to the confidence score approach. By definition, it is clear that $(p^*, \mathbf{\Pi}^*)$ is a feasible solution. All that is needed is to show the solution is optimal. Suppose not. We must have

$$\bar{r}_0 + p^* \left[\sum_{i=1}^K \mathbf{\Pi}'_i \bar{r}_{i,p} - \bar{r}_{0,p}\right] > \bar{r}_0 + p^* \left[\sum_{i=1}^K \mathbf{\Pi}_i^* \bar{r}_{i,p} - \bar{r}_{0,p}\right]$$

$$\sum_{i=1}^K \mathbf{\Pi}'_i \bar{r}_{i,p} > \sum_{i=1}^K \mathbf{\Pi}_i^* \bar{r}_{i,p}$$

But noting that $\mathbf{\Pi}'$ by definition is also a feasible solution to the problem C.7, this inequality implies that the objective function achieved by $\mathbf{\Pi}'$ is strictly larger than that achieved by one optimal solution to C.7. A contradiction. Hence, $(p^*, \mathbf{\Pi}^*)$ is one optimal solution.

Next we show that $\mathbf{\Pi}^*$ must follow the presented form. Since $\|\mathbf{\Pi}^*\|_0 \leq 2$, we can consider the cases separately.

(i) $\|\mathbf{\Pi}^*\|_0 = 1$: Assume $\mathbf{\Pi}_i^* \neq 0$. Then problem C.7 becomes

$$\max_{\mathbf{\Pi}_i \in \mathbb{R}^+} \mathbf{\Pi}_i \bar{r}_{i,p^*}$$

$$s.t. \ \mathbf{c}_i \mathbf{\Pi}_i \leq \frac{B}{p^*}, \mathbf{\Pi}_i \leq 1$$

Since the objective function is monotonely increasing w.r.t. $\mathbf{\Pi}_i$, we must have $\mathbf{\Pi}_i^* = \min\{\frac{B}{p^* \mathbf{c}}, 1\}$

(ii) $\|\mathbf{\Pi}^*\|_0 = 2$: Assume $\mathbf{\Pi}_i^* \neq 0, \mathbf{\Pi}_j^* \neq 0$. Then problem C.7 becomes

$$\max_{\mathbf{\Pi}_i \in \mathbb{R}^+, \mathbf{\Pi}_j \in \mathbb{R}^+} \mathbf{\Pi}_i \bar{r}_{i,p^*} + \mathbf{\Pi}_j \bar{r}_{j,p^*}$$

$$s.t. \ \mathbf{c}_i \mathbf{\Pi}_i + \mathbf{c}_j \mathbf{\Pi}_j \leq \frac{B}{p^*}, \mathbf{\Pi}_i + \mathbf{\Pi}_j \leq 1$$

As a linear programming, if it has a solution, then there must exist one solution on the corner point. Since $\mathbf{\Pi}_i^* \neq 0, \mathbf{\Pi}_j^* \neq 0$, the two constraints must be satisfied to achieve a corner point. The two constraints form a system of linear equations, and solving it gives $\mathbf{\Pi}_i^* = \frac{B/p - \mathbf{c}_j}{\mathbf{c}_i - \mathbf{c}_j}, \mathbf{\Pi}_j^* = \frac{\mathbf{c}_i - B/p}{\mathbf{c}_i - \mathbf{c}_j}$, which completes the proof. □

Now we are ready to prove Lemma 12. Recall that in Algorithm 2, we compute $(\mu_1, i_1) = \arg\max_{\mu \in [0,1], i \in [K]} \phi_i(\mu)$ and $(\mu_2, i_2, j_2) = \arg\max_{\mu \in [\frac{\beta}{\mathbf{c}_i}, \min\{\frac{\beta}{\mathbf{c}_j}, 1\}], i, j \in [K], \mathbf{c}_i > \mathbf{c}_j} \phi_{i,j}(\mu)$. If $\phi_{i_1}(\mu_1) \geq \phi_{i_2, j_2}(\mu_2)$, let $\rho = \mu_1$ and $\mathbf{\Pi} = \left[\mathbb{1}_{\mu_1 < \frac{\beta}{\mathbf{c}_{i_1}}} + \frac{\beta}{\mathbf{c}_i} \mathbb{1}_{\mu_1 \geq \frac{\beta}{\mathbf{c}_{i_1}}}\right] \mathbf{e}_{i_1}$. Otherwise, let $\rho = \mu_2$ and $\mathbf{\Pi} = \frac{\beta/\mu_2 - \mathbf{c}_{j_2}}{\mathbf{c}_{j_2} - \mathbf{c}_{j_2}} \mathbf{e}_{i_2} + \frac{\mathbf{c}_{i_2} - \beta/\mu_2}{\mathbf{c}_{i_2} - \mathbf{c}_{i_2}} \mathbf{e}_{j_2}$. Recall that $\phi_i(\mu) \triangleq \bar{r}_{k,\ell}(\mathbf{1}_{K \times L}) + \min\{\frac{\beta}{\mathbf{c}_i}, \mu\} \tilde{\mathbf{r}}_i^{k,\ell}(\mu)$ and $\phi_{i,j}(\mu) \triangleq \bar{r}_{k,\ell}(\mathbf{1}_{K \times L}) + \frac{\beta - \mu \mathbf{c}_j}{\mathbf{c}_i - \mathbf{c}_j} \tilde{\mathbf{r}}_i^{k,\ell}(\mu) + \frac{\mu \mathbf{c}_i - \beta}{\mathbf{c}_i - \mathbf{c}_j} \tilde{\mathbf{r}}_j^{k,\ell}(\mu)$.

Let us consider the two cases separately.

(i): $\phi_{i_1}(\mu_1) \geq \phi_{i_2, j_2}(\mu_2)$, and thus $\rho = \mu_1$ and $\mathbf{\Pi} = \left[\mathbb{1}_{\mu_1 < \frac{\beta}{\mathbf{c}_{i_1}}} + \frac{\beta}{\mathbf{c}_i} \mathbb{1}_{\mu_1 \geq \frac{\beta}{\mathbf{c}_{i_1}}}\right] \mathbf{e}_{i_1}$.

According to Lemma 13, there exists a solution $\check{\rho}^*, \check{\mathbf{\Pi}}^*$ to the above problem, such that

- $\check{\mathbf{\Pi}}_i^* = 1$ for some $i$, and $\check{\mathbf{\Pi}}_k^* = 0$, for all $k \neq i$

- $\check{\mathbf{\Pi}}_i^* = \frac{\beta}{\rho \mathbf{c}_i}$ for some $i$, and $\check{\mathbf{\Pi}}_k^* = 0$, for all $k \neq i$

- $\check{\mathbf{\Pi}}_i^* = \frac{\beta/\rho - \mathbf{c}_j}{\mathbf{c}_i - \mathbf{c}_j}, \check{\mathbf{\Pi}}_j^* = \frac{\mathbf{c}_i - \beta/\rho}{\mathbf{c}_i - \mathbf{c}_j}$, for some distinct $i, j$, and $\check{\mathbf{\Pi}}_k^* = 0$, for all $k \neq i, j$

If the first or second condition happens, the objective then becomes $\phi_i(\check{\rho}^*)$. If the third condition happens, then the objective becomes $\phi_{i,j}(\check{\rho}^*)$. Since $\phi_{i_1}(\mu_1) \geq \phi_{i_2, j_2}(\mu_2)$, we must have $\phi_i(\check{\rho}^*) \geq \phi_{i,j}(\check{\rho}^*)$ and thus it must be either first or second condition. By construction of $\mu_1$, we must have $\mu_1 = \hat{\rho}^*$. If $\check{\rho}^* = \mu_1 < \frac{\beta}{\mathbf{c}_{i_1}}$, i.e., $\frac{\beta}{\mathbf{c}_{i_1} \check{\rho}^*} > 1$, and thus second case cannot happen, and it has to be the first case and thus $\check{\mathbf{\Pi}}_{i_1}^* = 1$. By definition, we also have $\mathbf{\Pi} = \mathbf{e}_{i_1}$. And thus, we have $\check{\mathbf{\Pi}}^* = \mathbf{\Pi}$. If $\check{\rho}^* = \mu_1 \geq \frac{\beta}{\mathbf{c}_{i_1}}$, i.e., $\frac{\beta}{\mathbf{c}_{i_1} \check{\rho}^*} \leq 1$, then the second case must happen. Thus, we must have $\check{\mathbf{\Pi}}_{i_1}^* = \frac{\beta}{\rho \mathbf{c}}$. Meanwhile, by definition, we have $\mathbf{\Pi} = \frac{\beta}{\mathbf{c}_i \rho} \mathbf{e}_{i_1} = \check{\mathbf{\Pi}}^*$.

(ii): $\phi_{i_1}(\mu_1) \geq \phi_{i_2, j_2}(\mu_2)$, and thus $\rho = \mu_2$ and $\mathbf{\Pi} = \frac{\beta/\mu_2 - \mathbf{c}_{j_2}}{\mathbf{c}_{j_2} - \mathbf{c}_{j_2}} \mathbf{e}_{i_2} + \frac{\mathbf{c}_{i_2} - \beta/\mu_2}{\mathbf{c}_{i_2} - \mathbf{c}_{i_2}} \mathbf{e}_{j_2}$. We can use a similar argument to show that $\mathbf{\Pi} = \check{\mathbf{\Pi}}^*$.

That is to say, no matter which case we are in, the optimal solution is always returned. □

**Lemma 14.** *The function $\hat{h}_{k,\ell}(\beta)$ is Lipschitz continuous with constant $O(\gamma)$ for $\beta \geq 0$.*

*Proof.* Let us use $\phi_{k,\ell}()$ to denote $\hat{h}_{k,\ell}()$ for notation simplification. Consider $\beta$ and $\beta + \Delta$, and our goal is to bound $\phi_{k,\ell}(\beta + \Delta) - \phi_{k,\ell}(\beta)$. Let $\rho^{\beta+\Delta}, \mathbf{\Pi}^{\beta+\Delta}$ be the corresponding solution to $\beta + \Delta$, i.e., the solution to

$$\max_{\rho, \mathbf{\Pi} \in \Omega_2} \bar{\mathbf{r}}_{k,\ell}(\mathbf{1}_{K \times L}) + \rho \mathbf{\Pi}^T \cdot \tilde{\mathbf{r}}^{k,\ell}(\rho)$$

$$s.t. \ \rho(\mathbf{\Pi} - \mathbf{\Pi} \odot \mathbf{e}_k)^T \mathbf{c} \leq \beta + \Delta.$$

Let $\rho' = \frac{\beta}{\beta+\Delta}\rho^{\beta+\Delta}$. It is clear that $\rho', \mathbf{\Pi}^{\beta+\Delta}$ is one solution to

$$\max_{\rho, \mathbf{\Pi} \in \Omega_2} \bar{\mathbf{r}}_{k,\ell}(\mathbf{1}_{K \times L}) + \rho \mathbf{\Pi}^T \cdot \tilde{\mathbf{r}}^{k,\ell}(\rho)$$

$$s.t. \ \rho(\mathbf{\Pi} - \mathbf{\Pi} \odot \mathbf{e}_k)^T \mathbf{c} \leq \beta.$$

Thus, $\bar{\mathbf{r}}_{k,\ell}(\mathbf{1}_{K \times L}) + \rho' \mathbf{\Pi}^{(\beta+\Delta)T} \cdot \tilde{\mathbf{r}}^{k,\ell}(\rho')$ must be smaller or equal to $\phi_{k,\ell}(\beta)$, which is the optimal solution. Thus we must have

$$\phi_{k,\ell}(\beta + \Delta) - \phi_{k,\ell}(\beta) \leq \phi_{k,\ell}(\beta + \Delta) - \bar{\mathbf{r}}_{k,\ell}(\mathbf{1}_{K \times L}) - \rho' \mathbf{\Pi}^{(\beta+\Delta)T} \cdot \tilde{\mathbf{r}}^{k,\ell}(\rho')$$

Note that by definition,

$$\phi_{k,\ell}(\beta + \Delta) = \bar{\mathbf{r}}_{k,\ell}(\mathbf{1}_{K \times L}) + \rho^{\beta+\Delta}\mathbf{\Pi}^{(\beta+\Delta)T} \cdot \tilde{\mathbf{r}}^{k,\ell}(\rho^{\beta+\Delta})$$

The above inequality becomes

$$\phi_{k,\ell}(\beta + \Delta) - \phi_{k,\ell}(\beta) \leq \rho^{\beta+\Delta}\mathbf{\Pi}^{(\beta+\Delta)T} \cdot \tilde{\mathbf{r}}^{k,\ell}(\rho^{\beta+\Delta}) - \rho' \mathbf{\Pi}^{(\beta+\Delta)T} \cdot \tilde{\mathbf{r}}^{k,\ell}(\rho')$$

$$= \rho^{\beta+\Delta}\mathbf{\Pi}^{(\beta+\Delta)T} \cdot \tilde{\mathbf{r}}^{k,\ell}(\rho^{\beta+\Delta}) - \rho' \mathbf{\Pi}^{(\beta+\Delta)T} \cdot \tilde{\mathbf{r}}^{k,\ell}(\rho^{\beta+\Delta})$$

$$+ \rho' \mathbf{\Pi}^{(\beta+\Delta)T} \cdot \tilde{\mathbf{r}}^{k,\ell}(\rho^{\beta+\Delta}) - \rho' \mathbf{\Pi}^{(\beta+\Delta)T} \cdot \tilde{\mathbf{r}}^{k,\ell}(\rho')$$

$$= \rho^{\beta+\Delta}\mathbf{\Pi}^{(\beta+\Delta)T} \cdot \tilde{\mathbf{r}}^{k,\ell}(\rho^{\beta+\Delta}) - \frac{\beta}{\beta+\Delta}\rho^{\beta+\Delta}\mathbf{\Pi}^{(\beta+\Delta)T} \cdot \tilde{\mathbf{r}}^{k,\ell}(\rho^{\beta+\Delta})$$

$$+ \rho' \mathbf{\Pi}^{(\beta+\Delta)T} \cdot \tilde{\mathbf{r}}^{k,\ell}(\rho^{\beta+\Delta}) - \rho' \mathbf{\Pi}^{(\beta+\Delta)T} \cdot \tilde{\mathbf{r}}^{k,\ell}(\rho')$$

$$= \frac{\Delta}{\beta+\Delta}\rho^{\beta+\Delta}\mathbf{\Pi}^{(\beta+\Delta)T} \cdot \tilde{\mathbf{r}}^{k,\ell}(\rho^{\beta+\Delta})$$

$$+ \rho' \mathbf{\Pi}^{(\beta+\Delta)T} \cdot \tilde{\mathbf{r}}^{k,\ell}(\rho^{\beta+\Delta}) - \rho' \mathbf{\Pi}^{(\beta+\Delta)T} \cdot \tilde{\mathbf{r}}^{k,\ell}(\rho')$$

where the first equality is by adding and subtracting $\rho' \mathbf{\Pi}^{(\beta+\Delta)T} \cdot \tilde{\mathbf{r}}^{k,\ell}(\rho^{\beta+\Delta})$, and the second equality is simply plugging in the value of $\rho'$. According to Lemma 13, $\mathbf{\Pi}^{\beta+\Delta}$ must be sparse.

(i) If $\mathbf{\Pi}_k^{\beta+\Delta} = 1$, then only the base service ($k$th service) is used when budget is $\beta + \Delta$ When the budget becomes smaller, i.e., becomes $\beta$, it is always possible to always use the base service. Hence, we must have $\phi_{k,\ell}(\beta + \Delta) - \phi_{k,\ell}(\beta) = 0$.

(ii) Otherwise, since $\|\mathbf{\Pi}^{\beta+\Delta}\| \leq 2$, there are at most two elements in $\mathbf{\Pi}^{\beta+\Delta}$ that are not zeros. Let $k_1, k_2 \neq k$ denote the indexes. Then the constraint gives

$$\rho^{\beta+\Delta}\mathbf{\Pi}_{k_1}^{\beta+\Delta}\mathbf{c}_{k_1} + \rho^{\beta+\Delta}\mathbf{\Pi}_{k_2}^{\beta+\Delta}\mathbf{c}_{k_2} \leq \beta + \Delta$$

$$(\rho^{\beta+\Delta}\mathbf{\Pi}_{k_1}^{\beta+\Delta} + \rho^{\beta+\Delta}\mathbf{\Pi}_{k_2}^{\beta+\Delta})\min_{j \neq k}\mathbf{c}_j \leq \rho^{\beta+\Delta}\mathbf{\Pi}_{k_1}^{\beta+\Delta}\mathbf{c}_{k_1} + \rho^{\beta+\Delta}\mathbf{\Pi}_{k_2}^{\beta+\Delta}\mathbf{c}_{k_2} \leq \beta + \Delta$$

That is to say,

$$\rho^{\beta+\Delta}(\mathbf{\Pi}_{k_1}^{\beta+\Delta} + \mathbf{\Pi}_{k_2}^{\beta+\Delta}) \leq (\beta + \Delta)/(\min_{j \neq k}\mathbf{c}_j)$$

Thus we have

$$\frac{\Delta}{\beta+\Delta}\rho^{\beta+\Delta}\mathbf{\Pi}^{(\beta+\Delta)T} \cdot \tilde{\mathbf{r}}^{k,\ell}(\rho^{\beta+\Delta}) = \frac{\Delta}{\beta+\Delta}\rho^{\beta+\Delta}(\mathbf{\Pi}_{k_1}^{(\beta+\Delta)T} \cdot \tilde{\mathbf{r}}_{k_1}^{k,\ell}(\rho^{\beta+\Delta}) + \mathbf{\Pi}_{k_1}^{(\beta+\Delta)T} \cdot \tilde{\mathbf{r}}_{k_1}^{k,\ell}(\rho^{\beta+\Delta}))$$

$$\leq \frac{\Delta}{\beta+\Delta}\rho^{\beta+\Delta}(\mathbf{\Pi}_{k_1}^{(\beta+\Delta)T} + \mathbf{\Pi}_{k_1}^{(\beta+\Delta)T})$$

$$\leq \frac{\Delta}{\beta+\Delta}(\beta + \Delta)/(\min_{j \neq k}\mathbf{c}_j)$$

$$= \frac{\Delta}{\min_{j \neq k}\mathbf{c}_j}$$

In addition, note that by assumption, $\tilde{\mathbf{r}}^{k,\ell}(\rho)$ is Lipschitz continuous with constant $\gamma$. Hence, we must have

$$\tilde{\mathbf{r}}_j^{k,\ell}(\rho') \geq \tilde{\mathbf{r}}_j^{k,\ell}(\rho^{\beta+\Delta}) - \gamma|(\rho' - \rho^{\beta+\Delta}|) = \tilde{\mathbf{r}}_j^{k,\ell}(\rho^{\beta+\Delta}) - \gamma\frac{\Delta}{\beta+\Delta}\rho^{\beta+\Delta}$$

And thus

$$\rho'\mathbf{\Pi}^{(\beta+\Delta)T} \cdot \tilde{\mathbf{r}}^{k,\ell}(\rho^{\beta+\Delta}) - \rho'\mathbf{\Pi}^{(\beta+\Delta)T} \cdot \tilde{\mathbf{r}}^{k,\ell}(\rho')$$

$$\leq \rho'\mathbf{\Pi}^{(\beta+\Delta)T} \cdot \tilde{\mathbf{r}}^{k,\ell}(\rho^{\beta+\Delta}) - \rho'\mathbf{\Pi}^{(\beta+\Delta)T} \cdot \tilde{\mathbf{r}}^{k,\ell}(\rho^{\beta+\Delta}) + \rho'\mathbf{\Pi}^{(\beta+\Delta)T} \cdot \mathbf{1}\gamma\frac{\Delta}{\beta+\Delta}\rho^{\beta+\Delta}$$

$$= \rho'(\mathbf{\Pi}_{k_1}^{(\beta+\Delta)T} + \mathbf{\Pi}_{k_2}^{(\beta+\Delta)T})\gamma\frac{\Delta}{\beta+\Delta}\rho^{\beta+\Delta}$$

$$= \rho^{\beta+\Delta}\frac{\beta}{\beta+\Delta}(\mathbf{\Pi}_{k_1}^{(\beta+\Delta)T} + \mathbf{\Pi}_{k_2}^{(\beta+\Delta)T})\gamma\frac{\Delta}{\beta+\Delta}\rho^{\beta+\Delta}$$

Now plugging in

$$\rho^{\beta+\Delta}(\mathbf{\Pi}_{k_1}^{\beta+\Delta} + \mathbf{\Pi}_{k_2}^{\beta+\Delta}) \leq (\beta+\Delta)/(\min_{j \neq k}\mathbf{c}_j)$$

We can further have

$$\rho'\mathbf{\Pi}^{(\beta+\Delta)T} \cdot \tilde{\mathbf{r}}^{k,\ell}(\rho^{\beta+\Delta}) - \rho'\mathbf{\Pi}^{(\beta+\Delta)T} \cdot \tilde{\mathbf{r}}^{k,\ell}(\rho')$$

$$\leq \rho^{\beta+\Delta}\frac{\beta}{\beta+\Delta}(\mathbf{\Pi}_{k_1}^{(\beta+\Delta)T} + \mathbf{\Pi}_{k_2}^{(\beta+\Delta)T})\gamma\frac{\Delta}{\beta+\Delta}\rho^{\beta+\Delta}$$

$$\leq \frac{\beta}{\beta+\Delta}(\beta+\Delta)/(\min_{j \neq k}\mathbf{c}_j)\gamma\frac{\Delta}{\beta+\Delta}\rho^{\beta+\Delta}$$

$$\leq \Delta\gamma/(\min_{j \neq k}\mathbf{c}_j)$$

Combining it with

$$\frac{\Delta}{\beta+\Delta}\rho^{\beta+\Delta}\mathbf{\Pi}^{(\beta+\Delta)T} \cdot \tilde{\mathbf{r}}^{k,\ell}(\rho^{\beta+\Delta}) \leq \frac{\Delta}{\min_{j \neq k}\mathbf{c}_j}$$

we have

$$\phi_{k,\ell}(\beta+\Delta) - \phi_{k,\ell}(\beta) \leq \frac{\Delta}{\beta+\Delta}\rho^{\beta+\Delta}\mathbf{\Pi}^{(\beta+\Delta)T} \cdot \tilde{\mathbf{r}}^{k,\ell}(\rho^{\beta+\Delta})$$
$$+ \rho'\mathbf{\Pi}^{(\beta+\Delta)T} \cdot \tilde{\mathbf{r}}^{k,\ell}(\rho^{\beta+\Delta}) - \rho'\mathbf{\Pi}^{(\beta+\Delta)T} \cdot \tilde{\mathbf{r}}^{k,\ell}(\rho')$$
$$\leq \Delta/(\min_{j \neq k}\mathbf{c}_j) + \Delta\gamma/(\min_{j \neq k}\mathbf{c}_j)$$
$$= \frac{\Delta(1+\gamma)}{\min_{j \neq k}\mathbf{c}_j}$$

Thus, no matter which case, we always have

$$\phi_{k,\ell}(\beta+\Delta) - \phi_{k,\ell}(\beta) \leq \frac{1+\gamma}{\min_{j \neq k}\mathbf{c}_j} \cdot \Delta$$

In addition, since $\phi_{k,\ell}(\beta)$ must be monotone, we have

$$\phi_{k,\ell}(\beta+\Delta) - \phi_{k,\ell}(\beta) \geq 0 \geq -\frac{1+\gamma}{\min_{j \neq k}\mathbf{c}_j} \cdot \Delta$$

That is to say, $\phi_{k,\ell}(\beta)$ is Lipschitz continuous with constant $\frac{1+\gamma}{\min_{j \neq k}\mathbf{c}_j}$, which finishes the proof. $\square$

Now we are ready to prove Lemma 11. By definition, there must exist a $\boldsymbol{\lambda}'$, such that $g_i'(b') = \sum_{\ell=1}^{L} \hat{\mathbf{A}}_{i,\ell}\hat{h}_{k,\ell}(\boldsymbol{\lambda}_\ell'(b' - \mathbf{c}_i))$. Let $\Lambda_M = \{\boldsymbol{\lambda} \in \mathbb{R}^L | \boldsymbol{\lambda} \geq 0, \mathbf{1}^T\boldsymbol{\lambda} = 1, \boldsymbol{\lambda}_\ell M \in [M] \cup \{0\}\}$. Then there must exists a $\hat{\boldsymbol{\lambda}} \in \Lambda_M$ such that $|\boldsymbol{\lambda}_\ell' - \hat{\boldsymbol{\lambda}}_\ell| \leq \frac{1}{M}$. By Lemma 14, we have $|\hat{h}_{k,\ell}(\boldsymbol{\lambda}_\ell'(b' - $

$\mathbf{c}_i))) - \hat{h}_{k,\ell}(\hat{\boldsymbol{\lambda}}_\ell(b' - \mathbf{c}_i))| \leq O(\frac{\gamma}{M})$. Note that $\hat{\mathbf{A}}$ is empirical probability matrix, by construction, $\sum_\ell^L \hat{\mathbf{A}}_{\mathbf{i},\ell} = 1$ and each $\hat{\mathbf{A}}_{i,\ell}$ is non-negative. Thus, we must have $|\sum_{\ell=1}^L \hat{\mathbf{A}}_{\mathbf{i},\ell}\hat{h}_{k,\ell}(\hat{\boldsymbol{\lambda}}_\ell(b' - \mathbf{c}_i)) - g_i'(b')| = |\sum_{\ell=1}^L \hat{\mathbf{A}}_{i,\ell}\hat{h}_{k,\ell}(\hat{\boldsymbol{\lambda}}_\ell(b' - \mathbf{c}_i)) - \sum_{\ell=1}^L \hat{\mathbf{A}}_{i,\ell}\hat{h}_{k,\ell}(\boldsymbol{\lambda}_\ell'(b' - \mathbf{c}_i))| \leq O(\frac{\gamma}{M})$. On the other hand, by construction, $\hat{g}_i(b')$ produced by the subroutine to solve problem 3.3 is $\sum_{\ell=1}^L \hat{\mathbf{A}}_{i,\ell}\hat{h}_{k,\ell}(\beta_{t_\ell^*}) = \max_{t_1,t_2,\cdots,t_L} \sum_{\ell=1}^L \hat{\mathbf{A}}_{i,\ell}\hat{h}_{k,\ell}(\beta_{t_\ell}) = \max_{t_1,t_2,\cdots,t_L} \sum_{\ell=1}^L \hat{\mathbf{A}}_{i,\ell}\hat{h}_{k,\ell}(\frac{t_\ell^*}{M}(b' - \mathbf{c}_i)) \geq \sum_{\ell=1}^L \hat{\mathbf{A}}_{i,\ell}\hat{h}_{k,\ell}(\hat{\boldsymbol{\lambda}}_\ell(b' - \mathbf{c}_i))$. Combing this with $\sum_{\ell=1}^L \hat{\mathbf{A}}_{i,\ell}\hat{h}_{k,\ell}(\hat{\boldsymbol{\lambda}}_\ell(b' - \mathbf{c}_i)) - g_i'(b') \geq -O(\frac{\gamma}{M})$, we immediately obtain $\hat{g}_i(b') - g_i'(b') \geq -O(\frac{\gamma}{M})$. Since by definition $g_i'(b')$ must be the optimal solution and thus we have $\hat{g}_i(b') - g_i'(b') \leq 0$. Thus, we have $|\hat{g}_i(b') - g_i'(b')| \leq O(\frac{\gamma}{M})$. By Lemma 12, the produced solution to problem B.1 is exactly the optimal solution. That is to say, for the generated solution $\hat{\rho}^{i,\ell}(\beta_{t_\ell^*}), \hat{\boldsymbol{\Pi}}^{i,\ell}(\beta_{t_\ell^*})$, at most $\beta_{t_\ell^*}$ budget might be used. Since the total budget is $\sum_{\ell=1}^L \beta_{t_\ell^*} = b' - \mathbf{c}_i$, at most $b' - \mathbf{c}_i$ budget might be used. Calling the base service requires $\mathbf{c}_i$ cost, and thus the total cost is at most $b'$. As a result, we must have $\hat{\mathbb{E}}[\gamma^{s(i,b')(x)}] \leq b'$, which completes the proof. $\qquad\square$

**Lemma 15.** $|\hat{g}_i^{LI}(b') - g_i'(b')| \leq O(\frac{\gamma L}{M})$ for all $b'$ and $i$.

*Proof.* Let us consider three cases separately.

Case 1: $b' \leq \mathbf{c}_i$. By definition, $g_i'(b') = 0$. By construction, $\hat{g}_i^{LI}(b') = 0$, and thus $|\hat{g}_i^{LI}(b') - g_i'(b')| \leq O(\frac{\gamma}{M})$.

Case 2: $\theta_{m+1} \geq b' \geq \theta_m \geq \mathbf{c}_i$.

We first note that $g_i'(b')$ by definition, is

$$\max_{s=(\mathbf{e}_1),\mathbf{Q},\mathbf{P}\in S} \hat{\mathbb{E}}[r^s(x)|A_s^{[1]} = i]$$

$$s.t. \; \hat{\mathbb{E}}[\eta^s(x)] \leq b'$$

Abusing the notation a little bit, let us use $\mathbb{E}$ to denote $\hat{\mathbb{E}}$ for simplicity (as well as $\Pr$ for $\hat{\Pr}$). We can expand the objective function by

$$\mathbb{E}[r^s(x)|A_s^{[1]} = i]$$

$$= \sum_{\ell=1}^L \Pr[y_i(x) = \ell]\mathbb{E}[r^s(x)|A_s^{[1]} = i, y_i(x) = \ell]$$

$$= \sum_{\ell=1}^L \Pr[D_s = 0|A_s^{[1]} = i, y_i(x) = \ell]\Pr[y_i(x) = \ell]\mathbb{E}[r^s(x)|A_s^{[1]} = i, y_i(x) = \ell, D_s = 0]+$$

$$\sum_{\ell=1}^L \Pr[D_s = 1|A_s^{[1]} = i, y_i(x) = \ell]\Pr[y_i(x) = \ell]\mathbb{E}[r^s(x)|A_s^{[1]} = i, y_i(x) = \ell, D_s = 1]$$

$$= \sum_{\ell=1}^L \Pr[D_s = 0|A_s^{[1]} = i, y_i(x) = \ell]\Pr[y_i(x) = \ell]\mathbb{E}[r^i(x)|A_s^{[1]} = i, y_i(x) = \ell, D_s = 0]+$$

$$\sum_{\ell=1}^L \Pr[D_s = 1|A_s^{[1]} = i, y_i(x) = \ell]\Pr[y_i(x) = \ell]\Pr[A_s^{[2]} = j|A_s^{[1]} = i, y_i(x) = \ell, D_s = 1]\cdot$$

$$\mathbb{E}[r^s(x)|A_s^{[1]} = i, y_i(x) = \ell, D_s = 1, A_s^{[2]} = j]$$

$$= \sum_{\ell=1}^L \Pr[D_s = 0|A_s^{[1]} = i, y_i(x) = \ell]\Pr[y_i(x) = \ell]\mathbb{E}[r^i(x)|A_s^{[1]} = i, y_i(x) = \ell, D_s = 0]+$$

$$\sum_{\ell=1}^L \Pr[D_s = 1|A_s^{[1]} = i, y_i(x) = \ell]\Pr[y_i(x) = \ell]\mathbf{P}_{i,\ell,j}\cdot$$

$$\mathbb{E}[r^s(x)|A_s^{[1]} = i, y_i(x) = \ell, D_s = 1, A_s^{[2]} = j]$$

where all qualities are simply by applying the conditional expectation formula. That is to say, conditional on the quality score $\mathbf{Q}$, the objective function is a linear function over $\mathbf{P}$ where all coefficients are positive. Similarly, we can expand the budget constraint by

$$\mathbb{E}[\eta^s(x)|A_s^{[1]} = i]$$

$$= \sum_{\ell=1}^{L} \Pr[y_i(x) = \ell]\mathbb{E}[\eta^s(x)|A_s^{[1]} = i, y_i(x) = \ell]$$

$$= \sum_{\ell=1}^{L} \Pr[D_s = 0|A_s^{[1]} = i, y_i(x) = \ell]\Pr[y_i(x) = \ell]\mathbb{E}[\eta^s(x)|A_s^{[1]} = i, y_i(x) = \ell, D_s = 0]+$$

$$\sum_{\ell=1}^{L} \Pr[D_s = 1|A_s^{[1]} = i, y_i(x) = \ell]\Pr[y_i(x) = \ell]\mathbb{E}[\eta^s(x)|A_s^{[1]} = i, y_i(x) = \ell, D_s = 1]$$

$$= \sum_{\ell=1}^{L} \Pr[D_s = 0|A_s^{[1]} = i, y_i(x) = \ell]\Pr[y_i(x) = \ell]\mathbb{E}[\eta^i(x)|A_s^{[1]} = i, y_i(x) = \ell, D_s = 0]+$$

$$\sum_{\ell=1}^{L}\sum_{j=1}^{K} \Pr[D_s = 1|A_s^{[1]} = i, y_i(x) = \ell]\Pr[y_i(x) = \ell]\Pr[A_s^{[2]} = j|A_s^{[1]} = i, y_i(x) = \ell, D_s = 1]\cdot$$

$$\mathbb{E}[\eta^s(x)|A_s^{[1]} = i, y_i(x) = \ell, D_s = 1, A_s^{[2]} = j]$$

$$= \sum_{\ell=1}^{L} \Pr[D_s = 0|A_s^{[1]} = i, y_i(x) = \ell]\Pr[y_i(x) = \ell]\mathbf{c}_i+$$

$$\sum_{\ell=1}^{L}\sum_{j=1}^{K} \Pr[D_s = 1|A_s^{[1]} = i, y_i(x) = \ell]\Pr[y_i(x) = \ell]\mathbf{P}_{i,\ell,j}\mathbf{c}_j$$

which is also linear in $\mathbf{P}$ conditional on $\mathbf{Q}$. Let $(\mathbf{e}_i, \mathbf{Q}^*(b'), \mathbf{P}^*(b'))$ be the optimal solution that leads to $g_i'(b')$. Now let us consider

$$\max_{s=(\mathbf{e}_1, \mathbf{Q}^*(b'), \mathbf{P})\in S} \hat{\mathbb{E}}[r^s(x)|A_s^{[1]} = i]$$

$$s.t.\ \hat{\mathbb{E}}[\eta^s(x)] \leq b''$$

which is a linear programming over $\mathbf{P}$ which satisfies all conditions in Lemma 5. Let us denote its optimal value by $g_i'^{b'}(b'')$. When $b'' = b'$, its optimal value must be $g_i'(b')$, i,e., $g_i'('b') = g_i'^{b'}(b')$. By Lemma 5, we have $g_i^{b'}(\cdot)$ is Lipschitz continuous. In other words, we have $|g_i'^{b'}(b^1) - g_i'^{b'}(b^2)| \leq O(b_1 - b_2)$ for any $b_1, b_2, b'$. On the other hand, note that the optimal solution corresponding to $g_i'^{b'}(b'')$ is also a feasible solution to the original optimization without fixing $\mathbf{Q} = \mathbf{Q}^*(b')$. Hence, for any $b''$, we must have $g_i'(b') \geq g_i'^{b''}(b')$. Thus, for any $b_1 \leq b_2$, we have

$$g_i'(b^2) - g_i'(b^1) = g_i'(b^2) - g_i^{b^2}(b^2) + g_i^{b^2}(b^2) - g_i^{b^2}(b^1) + g_i^{b^2}(b^1) - g_i'(b^1)$$

$$= g_i^{b^2}(b^2) - g_i^{b^2}(b^1) + g_i^{b^2}(b^1) - g_i'(b^1)$$

$$\leq O(b_1 - b_2)$$

In addition, by definition $g_i'(b^2) - g_i'(b^1) \geq 0$. Hence, we have just shown that $|g_i'(b^2) - g_i'(b^1)| \leq O(b_1 - b_2)$, which implies $g'(b')$ is a Lipschitz continuous function. Lemma 11 implies that $|\hat{g}_{\theta_m} - g'(\theta_m)| \leq O(\frac{\gamma}{M})$ for every $m$. Now by Lemma 6, we obtain that

$$|\hat{g}^{LI}(b') - g'(b')| \leq O(\frac{\gamma}{M}) + O(\frac{1}{M})$$

Case 3: $\theta_m \geq b' \geq \mathbf{c}_i \geq \theta_{m-1}$. Exactly the same argument from case 2 can be applied, while noting that we use $\mathbf{c}_i$ as the interpolation point.

Thus, we have just proved that on three separate intervals, we have $|\hat{g}^{LI}(b') - g'(b')| \leq O(\frac{\gamma}{M}) + O(\frac{1}{M})$. Therefore, for any $b', i$, we must have $|\hat{g}^{LI}(b') - g'(b')| \leq O(\frac{\gamma}{M}) + O(\frac{1}{M})$, which completes the proof. $\square$

Now we are ready to prove Lemma 10.

Note that by definition, $s' \triangleq (\mathbf{p}^{[1]'}, \mathbf{Q}', \mathbf{P}^{[2]'})$ is the optimal solution to the empirical accuracy and cost joint optimization problem. By Lemma 1, $\mathbf{p}^{[1]'}$ should also be 2-sparse. Let $i_1'$ and $i_2'$ be the corresponding indexes of the nonzero components, $p_1', p_2'$ are the probability of using them as the base service, and $b_1', b_2'$ be the budget allocated to them in strategy $s'$. Then this must be the optimal solution to the master problem

$$\max_{(i_1, i_2, p_1, p_2, b_1, b_2) \in C} p_1 g_{i_1}'(b_1/p_1) + p_2 g_{i_2}'(b_2/p_2) \; s.t. b_1 + b_2 \leq b \tag{C.8}$$

On the other hand, due to the linear interpolation, the subroutine to solve master problem 3.2 in Algorithm 1 is effectively solving

$$\max_{(i_1, i_2, p_1, p_2, b_1, b_2) \in C} p_1 g_{i_1}^{LI}(b_1/p_1) + p_2 g_{i_2}^{LI}(b_2/p_2) \; s.t. b_1 + b_2 \leq b \tag{C.9}$$

and returns its optimal solution $\hat{i}_1, \hat{i}_2, \hat{p}_1, \hat{p}_2, \hat{b}_1, \hat{b}_2$. By Lemma 15, we have $|g_i'(b') - g_i^{LI}(b')| \leq O(\frac{\gamma L}{M})$ for all $b'$ and $i$, and thus for any $i_1, i_2, b_1, b_2, p_1, p_2 \in C$, we must have $|p_1 g_{i_1}'(b_1/p_1) + p_2 g_{i_2}'(b_2/p_2) - (p_1 g_{i_1}^{LI}(b_1/p_1) + p_2 g_{i_2}^{LI}(b_2/p_2))| \leq O(\frac{\gamma L}{M})$, since $p_1 + p_2 = 1$. Note that the constraints of the above two optimization are the same. Now we can apply Lemma 7, and obtain

$$\hat{p}_1 g_{\hat{i}_1}'(\hat{b}_1/\hat{p}_1) + \hat{p}_2 g_{\hat{i}_2}'(\hat{b}_2/\hat{p}_2) \geq p_1' g_{i_1'}'(b_1'/p_1') + p_2' g_{i_2'}'(b_2'/p_2') - O(\frac{\gamma L}{M}) \tag{C.10}$$

By definition, we have $\mathbb{E}[r^{s'}(x)] = p_1' g_{i_1'}'(b_1'/p_1') + p_2' g_{i_2'}'(b_2'/p_2')$, and thus the above simply becomes

$$\hat{p}_1 g_{\hat{i}_1}'(\hat{b}_1/\hat{p}_1) + \hat{p}_2 g_{\hat{i}_2}'(\hat{b}_2/\hat{p}_2) \geq \mathbb{E}[r^{s'}(x)] - O(\frac{\gamma L}{M}) \tag{C.11}$$

Next note that the final strategy is produced by calling subproblem 3.3 solver for $b' = \hat{b}_j/\hat{p}_j$ and $i = \hat{i}_j$, where $j = 1, 2$, and then aligning those two solutions. Thus, the empirical accuracy is simply $\mathbb{E}[r^{\hat{s}}(x)] = \hat{p}_1 \hat{g}_{\hat{i}_1}(\hat{b}_1/\hat{p}_1) + \hat{p}_2 \hat{g}_{\hat{i}_2}(\hat{b}_2/\hat{p}_2)$. By Lemma 11, we have

$$|\hat{p}_1 \hat{g}_{\hat{i}_1}(\hat{b}_1/\hat{p}_1) - \hat{p}_1 g_{\hat{i}_1}'(\hat{b}_1/\hat{p}_1)| \leq O(\frac{\gamma L}{M})$$

and

$$|\hat{p}_2 \hat{g}_{\hat{i}_2}(\hat{b}_2/\hat{p}_2) - \hat{p}_2 g_{\hat{i}_2}'(\hat{b}_1/\hat{p}_2)| \leq O(\frac{\gamma L}{M})$$

Adding those two terms we have

$$\hat{p}_1 \hat{g}_{\hat{i}_1}(\hat{b}_1/\hat{p}_1) - \hat{p}_1 g_{\hat{i}_1}'(\hat{b}_1/\hat{p}_1) + \hat{p}_2 \hat{g}_{\hat{i}_2}(\hat{b}_2/\hat{p}_2) - \hat{p}_2 g_{\hat{i}_2}'(\hat{b}_1/\hat{p}_2) \geq -O(\frac{\gamma L}{M})$$

That is to say,

$$\mathbb{E}[r^{\hat{s}}(x)] - \hat{p}_1 g_{\hat{i}_1}'(\hat{b}_1/\hat{p}_1) - \hat{p}_2 g_{\hat{i}_2}'(\hat{b}_1/\hat{p}_2) \geq -O(\frac{\gamma L}{M})$$

Adding the inequality C.11, we have

$$\mathbb{E}[r^{\hat{s}}(x)] - \mathbb{E}[r^{s'}(x)] \geq -O(\frac{\gamma L}{M})$$

which completes the proof. $\square$

Now let us prove Theorem 16, a slightly weaker version of Theorem 3.

**Theorem 16.** *Suppose $\mathbb{E}[r_i(x)|D_s = 0, A_s^{[1]} = i]$ is Lipschitz continuous with constant $\gamma$ w.r.t. each element in $\mathbf{Q}$. Given $N$ i.i.d. samples $\{y(x_i), \{(y_k(x_i), q_k(x_i))\}_{k=1}^K\}_{i=1}^N$, the computational cost of Algorithm 1 is $O\left(NMK^2 + K^3M^3L + M^LK^2\right)$. With probability $1 - \epsilon$, the produced strategy $\hat{s}$ satisfies $\mathbb{E}[r^{\hat{s}}(x)] - \mathbb{E}[r^{s^*}(x)] \geq -O\left(\sqrt{\frac{\log \epsilon + \log M + \log K + \log L}{N}} + \frac{\gamma L}{M}\right)$, and $\mathbb{E}[\gamma^{[\hat{s}]}(x, \mathbf{c})] \leq b + O\left(\sqrt{\frac{\log \epsilon + \log M + \log K + \log L}{N}}\right)$.*

*Proof.* There are three main parts: (i) the computational complexity, (ii) the accuracy drop, and (iii) the excessive cost. Let us handle them sequentially.

(i) Computational Complexity: Lemma 8 directly gives the computational complexity bound.

(ii) Accuracy Loss: By Lemma 9, with probability $1 - \epsilon$, we have

$$\mathbb{E}[\mathbb{1}_{\hat{y}^{\hat{s}}(x)=y(x)}] - \hat{\mathbb{E}}[\mathbb{1}_{\hat{y}^{\hat{s}}(x)=y(x)}] \geq -O\left(\sqrt{\frac{\log \epsilon + \log M + \log K + \log L}{N}} + \frac{\gamma}{M}\right)$$

$$\hat{\mathbb{E}}[\mathbb{1}_{\hat{y}^{s'}(x)=y(x)}] - \mathbb{E}[\mathbb{1}_{\hat{y}^{s'}(x)=y(x)}] \geq -O\left(\sqrt{\frac{\log \epsilon + \log M + \log K + \log L}{N}}\right)$$

and also

$$\mathbb{E}[\mathbb{1}_{\hat{y}^{s'}(x)=y(x)}] - \mathbb{E}[\mathbb{1}_{\hat{y}^{s^*}(x)=y(x)}] \geq -O\left(\sqrt{\frac{\log \epsilon + \log M + \log K + \log L}{N}} + +\frac{\gamma}{M}\right)$$

By Lemma 10, we have

$$\hat{\mathbb{E}}[\mathbb{1}_{\hat{y}^{\hat{s}}(x)=y(x)}] - \hat{\mathbb{E}}[\mathbb{1}_{\hat{y}^{s'}(x)=y(x)}] \geq -O\left(\frac{\gamma}{M}\right)$$

Combining those four inequalities, we have

$$
\begin{aligned}
&\mathbb{E}[\mathbb{1}_{\hat{y}^{\hat{s}}(x)=y(x)}] - \mathbb{E}[\mathbb{1}_{\hat{y}^{s^*}(x)=y(x)}] \\
=& \mathbb{E}[\mathbb{1}_{\hat{y}^{\hat{s}}(x)=y(x)}] - \hat{\mathbb{E}}[\mathbb{1}_{\hat{y}^{\hat{s}}(x)=y(x)}] + \hat{\mathbb{E}}[\mathbb{1}_{\hat{y}^{\hat{s}}(x)=y(x)}] - \hat{\mathbb{E}}[\mathbb{1}_{\hat{y}^{s'}(x)=y(x)}] \\
&+ \hat{\mathbb{E}}[\mathbb{1}_{\hat{y}^{s'}(x)=y(x)}] - \mathbb{E}[\mathbb{1}_{\hat{y}^{s'}(x)=y(x)}] + \mathbb{E}[\mathbb{1}_{\hat{y}^{s'}(x)=y(x)}] - \mathbb{E}[\mathbb{1}_{\hat{y}^{s^*}(x)=y(x)}] \\
\geq& -O\left(\sqrt{\frac{\log \epsilon + \log M + \log K + \log L}{N}}\right) - O\left(\frac{\gamma L}{M}\right) \\
&- O\left(\sqrt{\frac{\log \epsilon + \log M + \log K + \log L}{N}}\right) - O\left(\sqrt{\frac{\log \epsilon + \log M + \log K + \log L}{N}}\right) \\
\geq& -O\left(\sqrt{\frac{\log \epsilon + \log M + \log K + \log L}{N}} + \frac{\gamma L}{M}\right)
\end{aligned}
$$

(iii) Excessive Cost: Similar to (ii), by Lemma 9, with probability $1 - \epsilon$, we have

$$\mathbb{E}[\tau^{[\hat{s}]}(x, \mathbf{c})] - \hat{\mathbb{E}}[\tau^{[\hat{s}]}(x, \mathbf{c})] \leq O\left(\sqrt{\frac{\log \epsilon + \log K + \log L}{N}}\right)$$

$$\hat{\mathbb{E}}[\tau^{[s']}(x, \mathbf{c})] - \mathbb{E}[\tau^{[s']}(x, \mathbf{c})] \leq O\left(\sqrt{\frac{\log \epsilon + \log K + \log L}{N}}\right)$$

and also

$$\mathbb{E}[\tau^{[s']}(x, \mathbf{c})] - \mathbb{E}[\tau^{[s^*]}(x, \mathbf{c})] \leq O\left(\sqrt{\frac{\log \epsilon + \log K + \log L}{N}}\right)$$

By Lemma 10, we have

$$\hat{\mathbb{E}}[\tau^{[\hat{s}]}(x, \mathbf{c})] - \hat{\mathbb{E}}[\tau^{[s']}(x, \mathbf{c})] \le 0$$

Combining those four inequalities, we have

$$\mathbb{E}[\tau^{[\hat{s}]}(x, \mathbf{c})] - \mathbb{E}[\tau^{[s^*]}(x, \mathbf{c})]$$
$$= \mathbb{E}[\tau^{[\hat{s}]}(x, \mathbf{c})] - \hat{\mathbb{E}}[\tau^{[\hat{s}]}(x, \mathbf{c})] + \hat{\mathbb{E}}[\tau^{[\hat{s}]}(x, \mathbf{c})] - \hat{\mathbb{E}}[\tau^{[s']}(x, \mathbf{c})]$$
$$+ \hat{\mathbb{E}}[\tau^{[s']}(x, \mathbf{c})] - \mathbb{E}[\tau^{[s']}(x, \mathbf{c})] + \mathbb{E}[\tau^{[s']}(x, \mathbf{c})] - \mathbb{E}[\tau^{[s^*]}(x, \mathbf{c})]$$
$$\le O\left(\sqrt{\frac{\log \epsilon + \log K + \log L}{N}}\right) + 0$$
$$+ O\left(\sqrt{\frac{\log \epsilon + \log K + \log L}{N}}\right) + O\left(\sqrt{\frac{\log \epsilon + \log K + \log L}{N}}\right)$$
$$\le O\left(\sqrt{\frac{\log \epsilon + \log K + \log L}{N}}\right)$$

which completes the proof. $\qquad\square$

**Lemma 17.** *Let* $s^{\Delta b} \triangleq \arg\max_{s \in S} \mathbb{E}[r^s(x)]$ *s.t.* $\mathbb{E}[\gamma^{[s]}(x)] \le b - \Delta b$. *If* $b - \Delta b \ge 0$, *then* $\mathbb{E}[r^{s^{\Delta b}}(x)] - \mathbb{E}[r^{s^*}(x)] \ge -O(\Delta b)$.

*Proof.* We simply need to show that $\mathbb{E}[r^{s^{\Delta b}}(x)]$ is Lipschitz continuous in $\Delta b$. To see this, let us expand $s^{\Delta b} = (\mathbf{p}^{\Delta b}, \mathbf{Q}^{\Delta b}, \mathbf{P}^{\Delta b})$ and consider the following optimization problem

$$\max_{s = (\mathbf{p}^0, \mathbf{Q}^{\Delta b}, \mathbf{P}) \in S} \mathbb{E}[r^s(x)] \text{ s.t. } \mathbb{E}[\gamma^{[s]}(x)] \le a. \tag{C.12}$$

By law of total expectation, we have

$$\mathbb{E}[r^s(x)] = \sum_{i=1}^{K} \Pr[A_s^{[1]} = i] \mathbb{E}[r^s(x) | A_s^{[1]} = i]$$

And we can further expand the conditional expectation by

$$\mathbb{E}[r^s(x) | A_s^{[1]} = i]$$
$$= \sum_{\ell=1}^{L} \Pr[y_i(x) = \ell] \mathbb{E}[r^s(x) | A_s^{[1]} = i, y_i(x) = \ell]$$
$$= \sum_{\ell=1}^{L} \Pr[D_s = 0 | A_s^{[1]} = i, y_i(x) = \ell] \Pr[y_i(x) = \ell] \mathbb{E}[r^s(x) | A_s^{[1]} = i, y_i(x) = \ell, D_s = 0] +$$
$$\sum_{\ell=1}^{L} \Pr[D_s = 1 | A_s^{[1]} = i, y_i(x) = \ell] \Pr[y_i(x) = \ell] \mathbb{E}[r^s(x) | A_s^{[1]} = i, y_i(x) = \ell, D_s = 1]$$

Note that

$$\mathbb{E}[r^s(x) | A_s^{[1]} = i, y_i(x) = \ell, D_s = 0] = \mathbb{E}[r^i(x) | A_s^{[1]} = i, y_i(x) = \ell, D_s = 0]$$

and

$$\mathbb{E}[r^s(x) | A_s^{[1]} = i, y_i(x) = \ell, D_s = 1]$$
$$= \sum_{j=1}^{K} \Pr[A_s^{[2]} = j | A_s^{[1]} = i, y_i(x) = \ell, D_s = 1] \mathbb{E}[r^s(x) | A_s^{[1]} = i, y_i(x) = \ell, D_s = 1, A_s^{[2]} = j]$$
$$= \sum_{j=1}^{K} \mathbf{Q}_{i,\ell,j} \mathbb{E}[r^s(x) | A_s^{[1]} = i, y_i(x) = \ell, D_s = 1, A_s^{[2]} = j]$$

where all qualities are simply by applying the conditional expectation formula. That is to say, conditional on the quality score $\mathbf{Q}$, the objective function is a linear function over $\mathbf{P}$ where all coefficients are positive. Similarly, we can expand the budget constraint, which turns out to be also linear in $\mathbf{P}$ conditional on $\mathbf{Q}$.

Thus, problem C.12 is a linear programming in $\mathbf{P}$. Therefore, its optimal value, denoted by $F(a|\Delta_b)$, must also be Lipschitz continuous in $a$, according to Lemma 5. In other words, we have $|F(a_2|\Delta_b) - F(a_1|\Delta_b)| \leq O(|a_1 - a_2|)$ for any $a_1, a_2, \Delta b$. When $a = b - \Delta b$, its optimal value must be $\mathbb{E}[r^{s^{\Delta b}}(x)]$, i,e., $\mathbb{E}[r^{s^{\Delta b}}(x)] = F(b - \Delta b|\Delta_b)$. On the other hand, note that the optimal solution corresponding to $F(b - \Delta_b|0)$ is also a feasible solution to the original optimization without fixing $\mathbf{p} = \mathbf{p}^0, \mathbf{Q} = \mathbf{Q}^0$. Hence, we must have $\mathbb{E}[r^{s^{\Delta b}}(x)] \geq F(b - \Delta b|0)$ since the former is the optimal solution and the latter is only a feasible solution. Thus, we have

$$
\mathbb{E}[r^{s^0}(x)] - \mathbb{E}[r^{s^{\Delta b}}(x)]
$$
$$
= \mathbb{E}[r^{s^0}(x)] - F(b|0) + F(b|0) - F(b - \Delta b|0) + F(b - \Delta b|0) - \mathbb{E}[r^{s^{\Delta b}}(x)]
$$
$$
= F(b|0) - F(b - \Delta b|0) + F(b - \Delta b|0) - \mathbb{E}[r^{s^{\Delta b}}(x)]
$$
$$
\leq O(b - (b - \Delta b)) = O(\Delta b)
$$

Note that $\mathbb{E}[r^{s^0}(x)] = \mathbb{E}[r^{s^*}(x)]$, we have proved the statement. $\qquad\square$

Now we can relax Theorem 16 to Theorem 3 by slightly modifying the subroutines in Algorithm 1. First, we can compute the excessive part in the cost given in Lemma 9, denoted by $b_e$. Next, for all the subroutines in Algorithm 1, replace $b$ by $b - b_e$ whenever applicable. Thus, the produced solution with high probability has $\mathbb{E}[\gamma^{[s]}(x)] \leq b$, since we already remove the $b_e$ term. However, noting that by subtracting this $b_e$, we effective change the optimization problem by allowing a smaller budget, which is a conservative approach. Now by Lemma 17, this incurs at most $O(b_e)$ accuracy drop. By Lemma 9, $b_e = O(\sqrt{(\log \epsilon + \log K + \log L)/N})$, which is subsumed by the accuracy drop in Theorem 16, which finishes the proof. 17

$\qquad\square$

## D   Experimental Details

We provide missing experimental details here.

**Experimental Setup.**   All experiments were run on a machine with 20 Intel Xeon E5-2660 2.6 GHz cores, 160 GB RAM, and 200GB disk with Ubuntu 16.04 LTS as the OS. Our code is implemented in python 3.7.

**ML tasks and services.**   Recall that We focus on three main ML tasks, namely, facial emotion recognition (*FER*), sentiment analysis (*SA*), and speech to text (*STT*).

*FER* is a computer vision task, where give a face image, the goal is to give its emotion (such as happy or sad). For *FER*, we use 3 different ML cloud services, Google Vision [9], Microsoft Face (MS Face) [11], and Face++[6]. We also use a pretrained convolutional neural network (CNN) freely available from github [13]. Both Microsoft Face and Face++ APIs provide a numeric value in [0,1] as the quality score for their predictions, while Google API gives a value in five categories, namely, "very unlikely", "unlikely", "possible", "likely", and "very likely". We transform this categorical value into numerical value by linear interpolation, i.e., the five values correspond to 0.2, 0.4, 0.6, 0.8, 1, respectively.

*SA* is a natural language processing (NLP) task, where the goal is to predict if the attitude of a given text is positive or negative. For *SA*, the ML services used in the experiments are Google Natural Language (Google NLP) [7], Amazon Comprehend (AMZN Comp) [2], and Baidu Natural Language Processing (Baidu NLP) [3]. For English datasets, we use Vader [33], a rule-based sentiment analysis engine. For Chinese datasets, we use another rule-based sentiment analysis tool Bixin [4].

(a) Task: *FER*.  (b) Task: *SA*.  (c) Task: *STT*.

Figure 7: Cost per 10,000 queries of different ML APIs. GitHub refers to the CNN Model [13] in *FER*, Vader [16] and Bixin [4] in SA , and DeepSpeech [14] in *STT*.

*STT* is a speech recognition task where the goal is to transform an utterance into its corresponding text. for *STT*, we use three common APIs: Google Speech [8], Microsoft Speech (MS Speech) [12], and IBM speech [10]. a deepspeech model[14, 19] from github is also used. Given the returned text from a API, we determine the API's predicted label as the label with smallest edit distance to the returned text. For example, if IBM API produces "for" for a sample in AUDIOMNIST, then its label becomes "four", since all other numbers have larger distance from the predicted text "for".

**Datasets.**  The experiments were conducted on 12 datasets.  The first four datasets, FER+[20], RAFDB[39], EXPW[59], and AFFECTNET[42] are *FER* datasets.  The images in FER+ was originally from the FER dataset for the ICML 2013 Workshop on Challenges in Representation, and the label was recreated by crowdsourcing. We only use the testing portion of FER+, since the CNN model from github was pretrained on its training set. For RAFDB and AFFECTNET, we only use the images for basic emotions since commercial APIs cannot work for compound emotions. For EXPW, we use the true bounding box associated with the dataset to create aligned faces first, and only pick the images that are faces with confidence larger than 0.6.

For *SA*, we use four datasets, YELP [18], IMDB [41], SHOP [15], and WAIMAI [17]. YELP and IMDB are both English text datasets. YELP is from the YELP review challenge. Each review is associated with a rating from 1,2,3,4,5. We transform rating 1 and 2 into negative, and rating 4 and 5 into positive. Then we randomly select 10,000 positive and negative reviews, respectively. IMDB is already polarized and partitioned into training and testing parts; we use its testing part which has 25,000 images. SHOP and WAIMAI are two Chinese text datasets. SHOP contains polarized labels for reviews for various purchases (such as fruits, hotels, computers). WAIMAI is a dataset for polarized delivery reviews. We use all samples from SHOP and WAIMAI.

Finally, we use the other four datasets for *STT*, namely, DIGIT [5], AUDIOMNIST[21], COMMAND [52] and FLUENT [40].  Each utterance in DIGIT and AUDIOMNIST is a spoken digit (i.e., 0-9). The sampling rate is 8 kHz for DIGIT and 48 kHz for AUDIOMNIST. Each sample in COMMAND is a spoken command such as "go", "left", "right", "up", and "down", with a sampling rate of 16 kHz. In total, there are 30 commands and a few white noise utterances.  FLUENT is another dataset for speech command.  The commands in FLUENT are typically a phrase (e.g., "turn on the light" or "turn down the music"). There are in total 248 phrases, mapped to 31 unique labels. The sampling rate is also 16 kHz.

**GitHub Model Cost.**  We evaluate the inference time of all GitHub models on an Amazon EC2 t2.micro instance, which is $0.0116 per hour.  The CNN model needs at most 0.016 seconds per 480 x 480 grey image, Bixin and Vader require at most 0.005 seconds for each text with less than 300 words, and DeepSpeech takes at most 0.5 seconds for each less than 15 seconds utterance.  Hence, their equivalent price is $0.0005, $0.00016, and $0.016 per 10,000 data points. As shown in Figure 7, the services from GitHub are much less expensive than the commercial ML services.

**Case Study Details.**  For comparison purposes, we also evaluate the performance of a mixture of experts, a simple majority vote, and a simple cascade approach on FER+ dataset.

Figure 8: Confusion matrix annotated with overall accuracy and cost on FER+ testing. The y-axis corresponds to the true label and x-axis represents the predicted label. Each entry in a confusion matrix is the likelihood that its corresponding label in x-axis is predicted given the corresponding true label in y-axis. For example, the 0.87 in (i) means that for all surprise images, FrugalML correctly predicts 87% of them as surprise,

For the mixture of experts, we use softmax for the gating network, and linear model on the domain space for the feature generation. This results in a strategy that ends up with always calling the best expert Microsoft. For the simple majority vote, we first transforms each API's confidence score $q$ and predicted label $\ell$ into its probability vector $\mathbf{v} \in \mathbb{R}^L$, by $\mathbf{v}_\ell = q, \mathbf{v}_j = (1-q)/(L-1), j \neq \ell$. This can be viewed as that the API gives a distribution of all labels for the input data point. Assuming independence, we simply sum all APIs' distributions and then produce the label with highest estimated probability. We also use a simple majority vote, where we simply return the label on which most API agrees on. For example, if GitHub (CNN), Google, and Face++ all give a label "surprise", the no matter what Microsoft produces, we choose "surprise" as the label. We break ties randomly.

Figure 8 shows the confusion matrix of FrugalML, along with all ML services and the other approaches (namely, mixture of experts, simple cascade, (simple majority vote), and

Figure 9: Label distribution on dataset FER+. Most of the facial images are neutral and happy faces, and only a few are fear and disgust.

majority vote). Among all the four services, we first note that there is an accuracy disparity for different facial emotions. In fact, GitHub (CNN) gives the highest accuracy on anger images (0.73%), fear (0.81%), happy (0.90%) and sad (0.60%), Face++ is best at disgust emotion (0.60%) and surprise (0.85%), while Microsoft is best at neutral (90%). Meanwhile, GitHub (CNN) gives a poor performance for neutral images, Face++ can hardly tell the differences between fear and surprise, and Google has a hard time distinguishing between anger and disgust images. This implies bias (and thus strength and weakness) from each ML API, leading to opportunities for optimization. We would also like to note that such biases may be of independent interest and explored for fairness study in the future.

We notice that the mixture of expert approach has the same confusion matrix as the Microsoft API. This is because the simple mixture of experts simply learns to always use the Microsoft API. Noting that we use a simple linear gating on the raw image space, this probably implies that Microsoft API has the best performance on any subspace in the raw image space produced by any hyperplane. More complicated mixture of experts approaches may lead to better performance, but requires more training complexity. Again, unlike FrugalML, mixture of experts does not allow users to specify their own budget/accuracy constraints.

Simple cascade approach allows accuracy cost trade-offs. As shown in Figure 7(f), while reaching the same accuracy as the best commercial API (Microsoft), it only asks for half of the cost. In fact, simple cascade uses GitHub (CNN) and Microsoft as the base service and add-on service with a fixed threshold for all labels. As a result, compared to Microsoft API, the prediction accuracy of neutral images drops significantly, while the accuracy on all the other labels increases, and thus resulting in the same accuracy. FrugalML, also with half of the cost of Microsoft API, actually gives an accuracy (84%) even higher than that of Microsoft API (81%). In fact, FrugalML identifies that only a vert small portion of images are disgust, and thus slightly sacrifices the accuracy on disgust images to improve the accuracy on all the other images. Compared to the simple cascade approach in Figure 7 (f), FrugalML, as shown in Figure 7 (i), produces higher accuracy on all classes of images except disgust images. Compared to Microsoft API (Figure 7), FrugalML slightly hurts the accuracy on fear, sad, and neutral images, but significantly improve the accuracy on happy and other images. Note that the strategy learned by FrugalML depends on the data distribution. As shown in Figure 9, most images are neutral and happy, and thus a slight drop on neutral images is worthy in exchange of a large improvement on happy images. Depending on the training data distribution, FrugalML may have learned different strategies as well.

Finally we note that while (simple) majority vote gives a poor accuracy (80% in Figure 8 (g)), the majority vote approach does lead to an accuracy (82%) higher than Microsoft API, although it is still lower than FrugalML's accuracy (84%). In addition, ensemble methods like majority vote need access to all ML APIs, and thus requires a cost of 30$, which is 5 times as large as the cost of FrugalML. Hence, they may not help reduce the cost effectively.

**Commercial API Only Study.** Furthermore, We evaluate FrugalML's performance using only MLaaS APIs excluding GH. To match the best API (Microsoft)'s performance, the learned FrugalML strategy always uses Face++ as the base service and occasionally calls Microsoft API (10$), leading to overall cost reduction of 17%. Alternatively, using the same cost target as the best API (10$), FrugalML achieves a 2% accuracy improvement.