[Reviews · NeurIPS 2020]

Review 1

Summary and Contributions: This paper helps users navigate the space of ML inference APIs, which have a large variation in terms of performance and cost. It proposes a framework that identifies the best strategy across APIs within a budget constraint, by learning the strengths and weaknesses of each API. The proposed framework is evaluated against real-world ML APIs from all the main cloud providers for a set of important classification tasks (facial recognition, speech recognition, etc.) on a large dataset. update based on authors's feedback: - I would like to thank the authors for the detailed feedback and addressing my main comment.

Strengths: This paper addresses an important real-world problem faced by MLaaS users. It presents a novel, formal framework that helps the MLaaS user transparently identify the best strategy for their particular ML classification task, within a budget constraint, across a large set of ML APIs offered by cloud vendors. The framework implements an efficient algorithm that solves this optimization problem, with provable guarantees. In an evaluation using real-world ML APIs from cloud vendors on a large dataset, the authors are able to show a significant cost reduction when using their method, while matching the accuracy of the best individual API, or an improvement over this accuracy at the same cost. Furthermore, the frameworks is robust in that it only requires a modest amount of training data to provide accurate strategies. The release of their annotated dataset is a nice addition to their work that benefits the research community.

Weaknesses: It seems that a lot of the cost benefits come from the fact that free models are available on Github and from the significantly lower cost per prediction on those models when ran on a cloud compute instance ("GitHub Model Cost" under D. in Supplementary Material) compared to the cost of commercial ML API calls. This GH cost does not include (i) the cost required for an expert to identify suitable open-source/free models alternative to MLaaS APIs, (ii) the cost required for setting up, maintaining and administering the GH-model-based inference "service" on general purpose compute resources. Moreover, for many users, especially enterprise-level ones, security and other features typically offered by production MLaaS are important, and hard to replicate in a custom deployment. For the above reasons, the comparison of MLaaS APIs with custom GH-model-based inference is not completely apples-to-apples. It would be nice to also see how FrugalML performs when limited to only using MLaaS services APIs, excluding GH.

Correctness: The claims and methodology used by the authors are correct.

Clarity: The authors did a great job in writing this paper. The paper is well written, easy to follow, and concise.

Relation to Prior Work: The authors clearly distinguish the contributions of their paper with respect to the related work.

Reproducibility: Yes

Additional Feedback:


Review 2

Summary and Contributions: The paper deals with learning to assess ML APIs in terms of predictive accuracy and quality score, where the latter depicts the confidence of the API. The gist of the learning problem is that each API has an assigned cost, which needs to be kept at a minimum as well. The authors define a novel strategy where a base API is chosen based on learnt conditional accuracies which might be overruled by an add-on API if the quality score is not sufficiently high. The optimal strategy is generated via solving a stated optimization problem. The empirical results on computer vision and NLP datasets with real-world APIs are promising in that the generated strategy reduces costs while achieving high predictive accuracies.

Strengths: * Very sensible application of ML to efficient API-reuse on the Web, which has an impact for the community. * Sufficiently novel: while there are other works on assessing service APIs using ML, the proposed approach also optimizes budget constraints and takes into account additional service quality data * Good empirical results with substantial cost savings while achieving high predictive accuracy on diverse datasets.

Weaknesses: * While the approach is sufficiently introduced, details on the implementation of conditional accuracy estimation and "quality of quality score" is missing

Correctness: The paper provides sufficient formalism for the problem at hand and provides substantial proofs for claims (in the supplementary section). The empirical methodology is adequate as well, although I have open questions about the provided information

Clarity: The paper is well-written - it nicely introduces the proposed method by explaining the required baselines and depicting the formal algorithmic details.

Relation to Prior Work: The related work covers essential areas. As the approach is strongly related to ensemble methods, one could additionally mention other seminal works (e.g. [1,2,3]) and not highlight mixture-of-experts alone, which, of course, is seminal as well. Especially model libraries seem highly related, as you also learn global conditional accuracies for tasks, if I am not mistaken. [1] Wolpert, D.H., 1992. Stacked generalization. Neural networks, 5(2), pp.241-259. [2] Breiman, L., 1996. Bagging predictors. Machine learning, 24(2), pp.123-140. [3] Caruana, R., Niculescu-Mizil, A., Crew, G. and Ksikes, A., 2004, July. Ensemble selection from libraries of models. In Proceedings of the twenty-first international conference on Machine learning (p. 18).

Reproducibility: Yes

Additional Feedback: The paper covers the interesting topic of efficient API-reuse and, in general, presents a solid method with promising results. My open questions mostly concern the evaluation of the approach. The result section is insightful, but am I missing how the conditional accuracies are estimated. From the paper I extract that you learn a model which performs instance-wise predictions, correct? How much left-out training data of the particular dataset (or other datasets) do you use for this? How easy/difficult is this task and do the results vary on the used datasets? I found a confusion matrix of the approaches in the supplementary material, but this does not answer these questions. The method is evaluated against MoE as well as weighted majority (WM), where it is mentioned that WM results in high costs due to the need for calling all APIs. I would be interested in now more about the empirical results on conditional accuracy estimation. For example, how would the results look like if only the best API would be called? Does this coincide with the results for MoE? How would the results look like if only the provided quality score is used? Following this thought, I am wondering why MoE on Facial Emotion Recognition always chooses the same API? How do you calculate the quality score for the Github CNN? And, for the datasets, how good is the quality score as conditional accuracy estimator? Lastly, why is the proposed protocol of a strategy constrained to a single add-on service? For the conducted evaluation this obviously suffices, but I can imagine that hard real-world datasets might have less optimal quality scores (plus hard to learn conditional accuracies). When would the runtime of the proposed approach be a practical issue? ### Update after author response ### I thank the authors for their response to the raised questions. I have read the other reviews as well as the author response and, as the questions have been answered well, I keep my positive score for the paper. I think it provides an original and well-executed approach.


Review 3

Summary and Contributions: This paper presents a framework that can learn the reliability of existing ML APIs and make use of them to classify data within a given budget. It requires an initial training set where all true labels and predictions from all APIs are available to learn the strategy. The classification for new data is done in a two-round style, where a base API is first picked, if the confidence is larger than a learned threshold, then the prediction will be returned, otherwise a second API will be selected based on the learned strategy, and its prediction will be returned. The authors provide theoretic guarantees that the best strategy can be learned and conducted extensive experiments to show the effectiveness of their method.

Strengths: This paper studies a practical problem, i.e. how to use ML APIs more accurately and cheaply. The proposed method is novel and well-motivated. It takes budget into account and is flexible enough for practitioners to balance accuracy and cost. I believe it has a wide audience in this community.

Weaknesses: It seems that the best strategy depends on the label distribution of the initial training set. Let’s say we are doing tweet sentiment analysis, and it’s easy to imagine that the best strategy when most tweets are positive would be different from when most tweets are negative. I believe that the proposed method will work when the real label distribution is invariant and the same as the initial dataset. But is the proposed method robust when they are not the same? What will happen if the label distribution is changing over time? Should we switch the strategy at some point?

Correctness: I have not checked the proof carefully, but I believe the experimental results are convincing. The proposed method is simple and easy to interpret, so it is very clear to me why it works. The authors have compared the proposed method with two oblations (GH and QS only) which supports their claim that each component in the proposed method helps improve the accuracy.

Clarity: This paper is well organized and easy to follow.

Relation to Prior Work: As far as I know, there is no baseline that works in the same setting. I am satisfied that the authors have compared their method with majority voting and a mixture-of-expert model. There is a line of work in combining crowdsourcing and active learning, where a model selects which worker to label which item and finally all collected labels are aggregated to infer the true labels. But usually crowdsourcing methods only assume there are discrete labels but no quality/confidence scores, as such scores are hard to collect from workers, so I don’t expect crowdsourcing methods are competitive in this scenario.

Reproducibility: Yes

Additional Feedback:


Review 4

Summary and Contributions: This paper proposes FrugalAI, a way to combine APIs and an open source API for cheaper ML. The paper is well written and the idea is simple and clear. The experiments I felt were reasonable. The one theory result about the method was solid but not surprising. The authors address an important problem.

Strengths: The idea proposed is simple, clear, compelling, and useful. The experiments are well done, I wish somehow one could understand better how these results would generalize to other types of datasets.

Weaknesses: The authors could have pushed a bit further on the comparison with other cascade architectures and a better understanding of how robust these results are would be nice. For example can a GAN mess up FrugalAI more so than a quality API.

Correctness: Yes as far as I can tell the empirical methodology seems correct.

Clarity: Yes.

Relation to Prior Work: I believe so but am not an expert.

Reproducibility: Yes

Additional Feedback:

[Author Response · NeurIPS 2020]

We thank all reviewers for their insightful and helpful feedback.

*R2: It would be nice to see how FrugalML performs when limited to only using MLaaS services APIs, excluding GH.*

A: We conducted an additional experiment on dataset FER+ using only MLaaS APIs excluding GH. To match the best API (Microsoft)'s performance, the learned FrugalML strategy always uses Face++ (5$) as the base service and occasionally calls Microsoft API (10$), leading to overall cost reduction of 17%. Alternatively, using the same cost target as the best API (10$), FrugalML achieves a 2% accuracy improvement. We'll add this to the revision.

*R3: As the approach is strongly related to ensemble methods, one could additionally mention other seminal works (e.g. [1,2,3]) and not highlight mixture-of-experts alone, which, of course, is seminal as well.*

A: Thanks for the suggestion; we will discuss these related works in the revision.

*From the paper I extract that you learn a model which performs instance-wise predictions, correct? How much left-out training data of the particular dataset (or other datasets) do you use for this? How easy/difficult is this task and do the results vary on the used datasets?*

A: Yes. Except for Figure 5, all experiments use 50% data for training and the remaining for evaluation. As Figure 5 shows, when the training sample size is larger than a few thousands, the performance becomes steady.

*How would the results look like if only the best API would be called? Does this coincide with the results for MoE? How would the results look like if only the provided quality score is used? Following this thought, I am wondering why MoE on Facial Emotion Recognition always chooses the same API? How do you calculate the quality score for the Github CNN? And, for the datasets, how good is the quality score as conditional accuracy estimator?*

A: The dot points in Figure 4 shows the accuracy if we only allow calling the best API. We are using the provided quality score from those APIs. We hypothesize that MoE chooses the same API because it relies on a linear model on the image features alone, on which the best API dominates. The GitHub CNN model is a VGG-19 variant which adopts a soft-max layer to compute the quality score. As shown in Figure R1 as below, the quality score has a high correlation to conditional accuracy.

Figure R1: Accuracy conditional on quality score on dataset FER+.

*Lastly, why is the proposed protocol of a strategy constrained to a single add-on service? When would the runtime of the proposed approach be a practical issue?*

A: This is mainly because using more services may increase the cost and the sample complexity for training. Allowing more add-on services would be an interesting direction of future extension.

*R4: I believe that the proposed method will work when the real label distribution is invariant and the same as the initial dataset. But is the proposed method robust when they are not the same? What will happen if the label distribution is changing over time? Should we switch the strategy at some point?*

A: In this paper we do assume that the label distribution does not change during inference. When the distribution has changed, the performance of the trained strategies might drop down and retraining or domain adaptation is needed for better performance. We will add a discussion on this for the revision.

*R6: The authors could have pushed a bit further on the comparison with other cascade architectures and a better understanding of how robust these results are would be nice. For example can a GAN mess up FrugalAI more so than a quality API.*

A: Thanks for the suggestion; we will discuss robustness further in the revision. Our experiments on diverse real datasets from different domains suggest that FrugalML is robust. Testing it on GANs is a great idea for future work. One advantage of FrugalML over model cascade is that standard cascade methods incur a fixed cost while the proposed FrugalML allows for different budget requirements.

[Meta-Review · NeurIPS 2020]

This paper addresses an important praactica problem arising in the use of machine learning APIs. Each API has some predictive accuracy and quality score (confidence) but also has an assigned cost, which we'd like to minimize. The authors give a method to accomplish this: a base API is chosen based on learnt conditional accuracies which might be overruled by an add-on API if the quality score is not sufficiently high. The optimal strategy is generated via solving a stated optimization problem. The paper presents some neat experiments with this method on computer vision and NLP datasets with real-world APIs. These appear promising in that the generated strategy reduces costs while still achieving high predictive accuracies. The reviewers were impressed with the solution to the practical problem and the high quality writing. I recommend this paper for acceptance.